# *Mcph1*, mutated in primary microcephaly, is also crucial for erythropoiesis

Yoann Vial [1,2], Jeannette Nardelli [3], Adeline A Bonnard[1,2], Justine Rousselot [2], Michèle Souyri[1], Pierre Gressens [3], Hélène Cavé [1,2] & Séverine Drunat [2,3]✉

## Abstract

Microcephaly is a common feature in inherited bone marrow failure syndromes, prompting investigations into shared pathways between neurogenesis and hematopoiesis. To understand this association, we studied the role of the microcephaly gene *Mcph1* in hematological development. Our research revealed that *Mcph1*-knockout mice exhibited congenital macrocytic anemia due to impaired terminal erythroid differentiation during fetal development. Anemia's cause is a failure to complete cell division, evident from tetraploid erythroid progenitors with DNA content exceeding 4n. Gene expression profiling demonstrated activation of the p53 pathway in *Mcph1*-deficient erythroid precursors, leading to overexpression of *Cdkn1a/p21*, a major mediator of p53-dependent cell cycle arrest. Surprisingly, fetal brain analysis revealed hypertrophied binucleated neuroprogenitors overexpressing p21 in *Mcph1*-knockout mice, indicating a shared pathophysiological mechanism underlying both erythroid and neurological defects. However, inactivating p53 in *Mcph1*$^{-/-}$ mice failed to reverse anemia and microcephaly, suggesting that p53 activation in *Mcph1*-deficient cells resulted from their proliferation defect rather than causing it. These findings shed new light on Mcph1's function in fetal hematopoietic development, emphasizing the impact of disrupted cell division on neurogenesis and erythropoiesis — a common limiting pathway.

**Keywords** Congenital Anemia; Cytokinesis; Mcph1; Neurogenesis; p53
**Subject Categories** Cell Cycle; Development

## Introduction

Primary microcephaly (PM) is a rare and genetically heterogeneous neurodevelopmental disorder defined by a head circumference at birth that is more than 3 DS smaller than average. PM is driven by a group of genes whose mutations result in reduced neuronal output during fetal development by altering the tight balance between neural progenitor cell (NPC) proliferation, differentiation, and death.

Such a balance is also crucial for the proper development of hematopoietic progenitor cells (HPCs) in which its disruption will lead to inherited bone marrow failure (iBMF). Whether the mechanism involved is an alteration in DNA damage repair (Fanconi anemia (FA)), telomere maintenance (dyskeratosis congenita (DC)), or ribosome biogenesis (Diamond-Blackfan anemia (DBA), Shwachman-Diamond syndrome (SBDS)), it is striking that microcephaly is a common feature in iBMF. Indeed, microcephaly is found in at least in 27%, 9%, 2%, and 1% of cases of FA, DC, DBA, and SBDS, respectively (Fiesco-Roa et al, 2019; Niewisch and Savage, 2019; Shimamura and Alter, 2010). This suggests that, during development, common molecular mechanisms are essential for neurogenesis and hematopoiesis.

The role of iBMF and DNA damage repair genes in brain development has been addressed in several studies (Sii-Felice et al, 2008; Zhou et al, 2020; Enriquez-Rios et al, 2017). Conversely, little is known about the role of microcephaly genes in hematopoietic development. We therefore questioned the possible role of MCPH1 in hematopoietic development. *MCPH1* or *Microcephalin*, is the first causative gene identified in microcephaly primary hereditary (MCPH), an autosomal recessive disorder, and is mutated in 10% of patients with MCPH (Kristofova et al, 2022). This gene is of particular interest for two reasons. First, public databases indicate that *Mcph1* is highly expressed in lymphocytes and erythroblasts. Second, it encodes microcephalin (MCPH1), a protein involved in many cellular functions that maintain genomic integrity during cell division, such as DNA repair, chromatin condensation, regulation of cell cycle checkpoints and telomere maintenance (Pulvers et al, 2015; Venkatesh and Suresh, 2014). Cellular defects in these functions are also involved in iBMF. Yet, it is not known whether MCPH1 plays a role in hematopoietic cells and, if so, what that role is. Using a well-characterized *Mcph1* knockout mouse model of microcephaly (Liang et al, 2010; Liu et al, 2021), we show here for the first time that loss of *Mcph1* results in impaired terminal differentiation of erythroid progenitors due to a p53-independent acytokinetic mitosis, leading to severe congenital anemia. We also show that these mechanisms are mirrored in neural progenitors, revealing similar vulnerabilities between neurogenesis and erythropoiesis.

[1]Université Paris Cité, Institut de Recherche Saint-Louis, Inserm UMR_S1131, F-75010 Paris, France. [2]Assistance Publique - Hôpitaux de Paris (AP-HP), Hôpital Robert Debré, Laboratoire de Génétique Moléculaire, F-75019 Paris, France. [3]Université Paris Cité, NeuroDiderot, Inserm, F-75019 Paris, France. ✉E-mail: severine.drunat@aphp.fr

# Results

## Mcph1 null mice exhibit severe congenital anemia with impaired terminal erythroid differentiation

To investigate a possible role of Mcph1 in hematopoiesis we generated $Mcph1^{-/-}$ and $Mcph1^{+/+}$ neonates by $Mcph1^{+/-}$ breeding. Of 146 live birth, 17% were $Mcph1^{-/-}$, 54% $Mcph1^{+/-}$, and 29% $Mcph1^{+/+}$. This distribution is significantly different from the expected distribution (Chi-square, p < 0.05) confirming that some of the $Mcph1^{-/-}$ mice do not reach the term (Liang et al, 2010). Furthermore, $Mcph1^{-/-}$ mice died within hours of birth, illustrating the severity of the phenotype.

At birth, $Mcph1^{-/-}$ mice recapitulated the phenotype of patients with bi-allelic *MCPH1* loss of function mutations, i.e., growth retardation and reduced head size (Liu et al, 2021). In addition, a striking pallor was observed in the knockout animals (KO) (Fig. 1A). When compared to their wild-type (WT) littermates, peripheral blood counts revealed a marked anemia in $Mcph1^{-/-}$ mice with a 1.6-fold reduction in hemoglobinemia (Hb) (90 vs 147 g/L in KO and WT mice, respectively; $p < 0.0001$) due to a drastic reduction of mature red blood cell (RBC) (2.7-fold reduction, 1.06 vs $2.82 \times 10^{12}$/L in KO and WT mice, respectively; $p < 0.0001$) (Fig. 1B,C). A reduction in B-lymphocytes was also noted (Appendix Fig. S2). Cytomorphologic examination of peripheral blood smears showed variation in the size (anisocytosis) and shape (poikilocytosis) of red blood cells as well as the presence of Howell–Jolly bodies and basophilic stippling in RBCs, and premature chromosome condensation in lymphocytes, a cellular phenotype also observed in *MCPH1* patients (Fig. 1D) (Trimborn et al, 2004). Flow cytometry analysis from Lin⁻ (B220⁻, CD4⁻, CD8a⁻, CD11b⁻ and Gr-1⁻) peripheral blood cells confirmed the huge decrease in circulating mature red blood cells (CD71$^{-/low}$, Ter119$^{High}$). This decrease was offset by an increase in the circulating erythroblast population (CD71$^{High}$, Ter119$^{High}$) (Fig. 1E). In addition, the median volume of mature circulating RBCs was higher and the width of the RBCs distribution broader in $Mcph1^{-/-}$ than in $Mcph1^{+/+}$ mice (Fig. 1F). These observations are consistent with $Mcph1^{-/-}$ mice having congenital macrocytic anemia with major dyserythropoiesis.

Noteworthy, animals with mono-allelic inactivation of *Mcph1* ($Mcph1^{+/-}$) showed normal brain development and no significant reduction in either Hb or RBC count, suggesting that *Mcph1* haploinsufficiency is not sufficient to impair either neurogenesis or erythropoiesis (Fig. 1B,C).

Macroscopic examination of hematopoietic tissues showed a drastic reduction in spleen size, far exceeding the reduction of overall body size in $Mcph1^{-/-}$ mice compared to their wild-type littermates (Fig. 1G). The liver size was unchanged. However, hematoxylin-eosin staining of neonatal liver and femur showed a decrease in the amount of hematopoietic cells in the liver and bone marrow in KO mice (Fig EV1A,B).

We then investigated whether the decreased RBC count in $Mcph1^{-/-}$ neonates was due to a block in erythroid differentiation. As the presence of neonatal anemia pointed to a prenatal defect, experiments were conducted on the liver, the main organ of fetal erythropoiesis. The proportion of erythroid progenitor subsets S0 to S5 was measured in live Lin⁻ cells obtained from the whole livers of newborn wild-type and null mice according to their CD71 and Ter119 immunophenotype (Fig. 1H). A drastic decrease in the proportion of the most differentiated cells (S4-S5 subsets; CD71$^{-/low}$, Ter119$^{High}$) ($p < 0.0001$) and an increase of cells in the S2 subset (CD71$^{High}$, Ter119$^{Med}$) ($p < 0.05$) was evidenced in $Mcph1^{-/-}$ erythroid cells (Fig. 1I,J). These data suggest that the neonatal anemia in KO mice is the consequence of a default in the progression of erythroid progenitors toward the terminal differentiation stages due to a defect occurring between the S2 and S4 stages.

## Mcph1 is necessary for efficient terminal erythroid differentiation during fetal development

To determine the developmental stage at which erythropoiesis fails, we examined embryos at 12.5 days of development (E12.5). Of 107 embryo, 25% were $Mcph1^{-/-}$, 46% $Mcph1^{+/-}$, and 29% $Mcph1^{+/+}$, which fits the expected distribution. This indicates that $Mcph1^{-/-}$ mice that do not reach term die after E12.5.

Anemia was already pronounced at E12.5, with $Mcph1^{-/-}$ embryos showing small livers, pallor, and faint vessels consistent with decreased RBC in bloodstream (Fig. 2A). Whole liver cell suspension counts confirmed a 2-fold reduction in fetal liver cellularity in $Mcph1^{-/-}$ fetuses compared to their wild-type counterparts ($p < 0.0001$) (Fig. 2B). At this stage of development, the fetal liver consists of 80–90% erythroid cells (Kina et al, 2000), suggesting that deficient erythropoiesis is likely to explain the reduction in liver size. Quantification of erythroid progenitor subpopulations in the fetal liver showed a significant decrease in the S3 subset whereas less differentiated subsets S1 and S2 increased (Fig. 2C,D). Taken together, these results show that disorders of erythroid differentiation arise as soon as hematopoiesis is established in the fetal liver.

## Lack of Mcph1 results in overexpression of the p53 target genes

To characterize the molecular defect underlying impaired erythropoiesis in $Mcph1^{-/-}$ embryos, RNAseq was performed on S0, S1, S2, and S3 subpopulations sorted from mouse fetal liver at E12.5 (Fig. 3A). As reported in healthy mice and humans, a decrease in *Gata2* expression, with increase in *Gata1* expression was observed in the S0 to S3 subtypes (Yu et al, 2020; Romano et al, 2020), validating our cell sorting (Fig. 3B).

Differential analysis by mouse genotype adjusted by differentiation stage and gender evidenced 78 genes differentially expressed between $Mcph1^{-/-}$ and $Mcph1^{+/+}$ embryos (p-adjust value < 0.05), of which, 55 were upregulated and 23 downregulated in $Mcph1^{-/-}$ erythroid precursors (Fig. 3C). Though binding sites for the master regulators of erythropoiesis, *Gata1* and *Tal1* are present in the *Mcph1* promoter, supporting a physiological link with erythropoiesis (Appendix Fig. S3), no change in the expression profile of the erythroid transcription program (Love et al, 2014b) was observed in KO mice when compared to WT mice. In contrast, Gene set enrichment analysis (GSEA) evidenced p53 pathway as the only biological process significantly enriched among Hallmark gene sets ($p = 0.001$, FDR = 0.022) (Fig. 3D). Eight genes encoding p53 targets were found among the 10 most overexpressed genes in *Mcph1* KO progenitors (Fischer, 2017), the most overexpressed of which was *Cdkn1a/p21*, the primary mediator of p53-dependent cell cycle arrest (Fig. 3E).

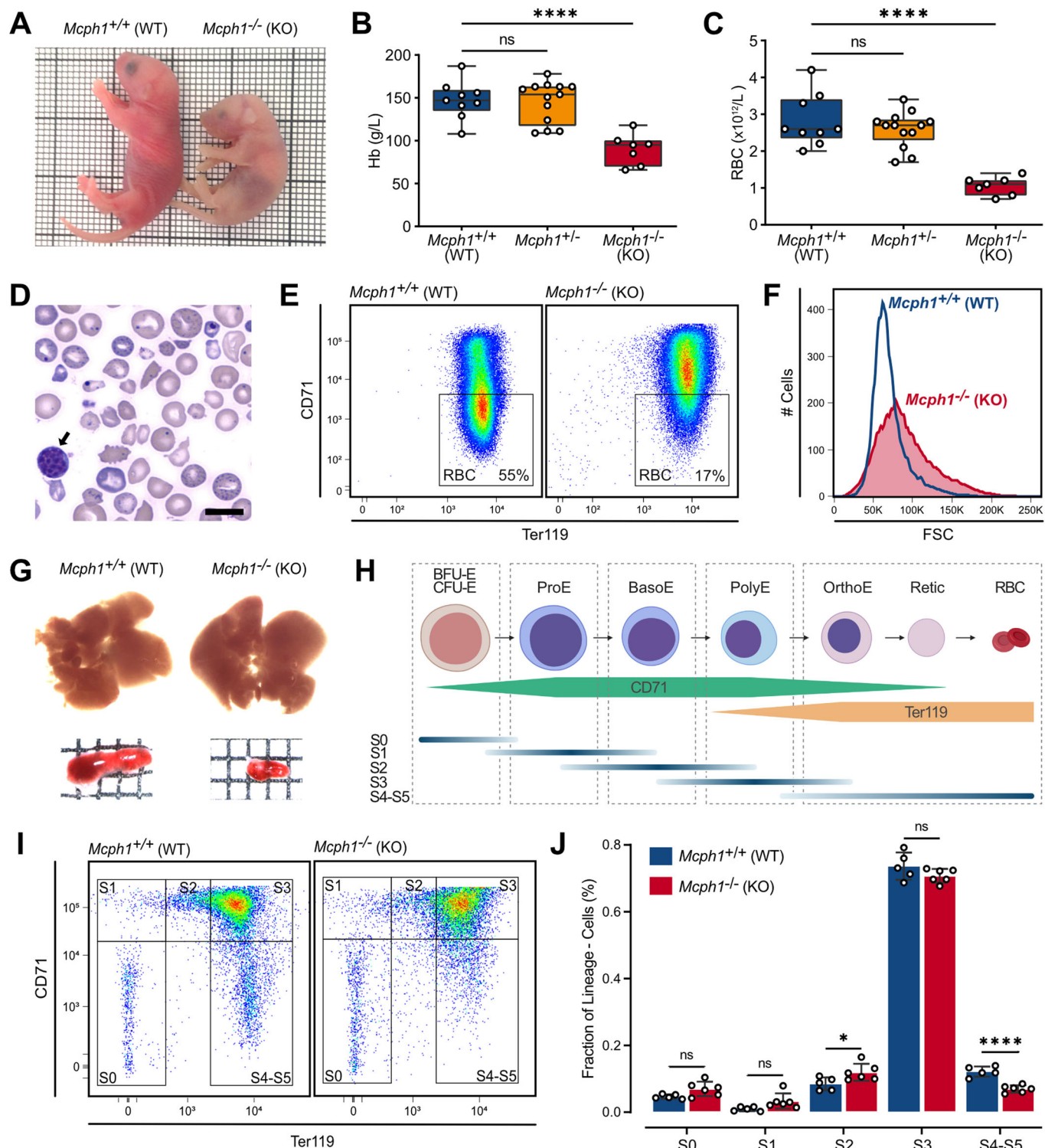

Quantification by digital droplet RT-PCR (ddRT-PCR) confirmed that *Cdkn1a/p21* transcript was overexpressed in each of the erythroid progenitor subtypes, with a gradual increase from 3.25-fold in S0 to 7.60-fold in S3 in *Mcph1⁻/⁻* embryos compared to *Mcph1⁺/⁺* embryos (Fig. 3F). Accordingly, p21, the protein encoded by *Cdkn1a*, demonstrated a 12-fold overexpression in the fetal liver of *Mcph1* KO embryos as compared to WT controls (Fig. 3G). This increased expression was also observed in immunofluorescence images of fetal liver sections (Fig. EV2A,B). Furthermore, given that the p53/p21 pathway plays a key role in triggering senescence in response to DNA damage, and that p16 is essential for the maintenance of senescence (Sheekey and Narita, 2023), we

◄  **Figure 1.  Total inactivation of *Mcph1* results in congenital anemia with major dyserythropoiesis in mice.**

Wild type (WT, *Mcph1*$^{+/+}$), heterozygous *Mcph1*$^{+/-}$, and knockout (KO, *Mcph1*$^{-/-}$) mice were examined at birth (day 0). **(A)** Photograph of newborn *Mcph1*$^{+/+}$ (left) and *Mcph1*$^{-/-}$ (right) mice. **(B, C)** Peripheral blood was collected from neonates of several litters to perform complete blood count. Boxes and whiskers represent Hemoglobinemia (Hb) in g/L and red blood cell (RBC) counts in ×10$^{12}$/L for *Mcph1*$^{+/+}$ (*n* = 9), *Mcph1*$^{+/-}$ (*n* = 13), and *Mcph1*$^{-/-}$ (*n* = 7) mice. Median, 25th, 75th percentiles, min and max. One-way anova, **** <0.0001, F(2,26) = 17.4 (Hb) and F(2,26) = 26.2 (RBC). **(D)** Cytological examination of May-Grünwald-Giemsa-stained blood smear from *Mcph1*$^{-/-}$ mice. Black arrow indicates lymphocyte with premature chromosome condensation. Scale bar = 10 μm. **(E)** CD71 and Ter119 expression was studied by flow cytometry on Lin⁻ cells collected from peripheral blood of newborn *Mcph1*$^{+/+}$ and *Mcph1*$^{-/-}$ mice. Mature RBC were defined as CD71⁻, Ter119$^{High}$ cells. **(F)** Size distribution of mature RBC was determined from forward scatter (FSC) on subsampled data (Median-FSCs are 66 K and 85 K for *Mcph1*$^{+/+}$ and *Mcph1*$^{-/-}$ mice, respectively). **(G)** Liver and spleen of *Mcph1*$^{+/+}$ and *Mcph1*$^{-/-}$ mice were collected for macroscopic examination. **(H)** Schematic representation of CD71 and Ter119 expression during terminal erythroid differentiation. Green and yellow boxes represent the expression level of CD71 and Ter119 in erythroid progenitors (BFU-E and CFU-E), nucleated erythroblasts (proerythroblasts (proE), basophilic erythroblasts (basoE), polychromatic erythroblasts (polyE) and orthochromatic erythroblasts (orthoE)), reticulocytes (Retic) and RBC. Gates based on CD71 and Ter119 expression on Lin⁻ cells allow to identify subpopulations S0 (CD71$^{-/low}$, Ter119⁻), S1 (CD71$^{High}$, Ter119⁻), S2 (CD71$^{High}$, Ter119$^{Med}$), S3 (CD71$^{High}$, Ter119$^{High}$), and S4-S5 (CD71$^{-/low}$, Ter119$^{High}$). Blue lines indicate the distribution of different cell types in the subpopulations defined by flow cytometry. Created with BioRender.com. **(I, J)** Livers of newborn *Mcph1*$^{+/+}$ (*n* = 5), *Mcph1*$^{-/-}$ (*n* = 6) mice from multiple litters were collected. The proportion of S0, S1, S2, S3, and S4-S5 subpopulations was determined by flow cytometry. Mean ± SD of 3 experiments. Multiple t-test for *p*-values (df = 9). * <0.05, **** <0.0001. Source data are available online for this figure.

performed immunofluorescence staining for p16 and γH2AX in fetal liver. We found no evidence of senescence or DNA damage, as evidenced by the absence of γH2AX foci, in KO fetal liver.

All these findings show that activation of the p53 pathway in *Mcph1* KO erythroid precursors results in the abnormal expression of *Cdkn1a*, a key player in cell cycle regulation.

## Mcph1 dysfunction leads to acytokinetic mitosis in the S3 subset

We thus investigated the cell cycle in *Mcph1*$^{-/-}$ erythroid progenitors isolated from newborn mouse liver. FACS analysis after cell labeling with both erythroid-specific antibodies (Ter119, CD71) and EdU-FxCycle showed no blockage of cell cycle progression in KO subsets S0 to S2 when compared with wild-type. However, a relative decrease in G1-phase cells compared with S and G2/M-phase cells was observed in KO mice (Fig. 4A). This was accompanied by an increase in sub-G1 cells in S0 and S2 KO progenitors. These observations are consistent with a regenerative process in which cells divide more at the cost of increased apoptosis. Strikingly, in subset S3, proper characterization of the different phases of the cell cycle was hampered by the presence of tetraploid cells that continue to progress through the cell cycle, leading to the formation of cells with 8n DNA content (Fig. 4B,C). Cell quantification by DNA content showed a relative increase in cells with a 4n (52.8 vs 40.5% in KO and WT mice, respectively) or >4n DNA content (12.6 vs 0.7% in KO and WT mice, respectively) versus cells with a 2n DNA content (28.1 vs 57.4% in KO and WT mice, respectively) in the *Mcph1*$^{-/-}$ S3 progenitor subset (Fig. 4D). Considering that the S3 population represents 70.7% of erythroblasts in KO animals, we can estimate that more than 8.9% of erythroblasts are tetraploid. Indeed, cytomorphologic examination of sorted newborn liver erythroid cells showed a high frequency of micronuclei or tetraploidy with two individualized nuclei (Fig. 4E,F). Such cells with two individualized nuclei suggest that cytokinesis failed (Gandarillas et al, 2018). Acytokinetic mitosis were also observed but at a much lower level in subsets S0 to S2 (Fig. 4D). Intriguingly, anillin staining, a protein localized in the contractile ring during cytokinesis, did not show any abnormalities in cells at the cytokinesis stage in KO fetal livers (Fig. EV3).

However, the presence of numerous polyploid erythroid progenitors strongly suggests that the congenital anemia observed in *Mcph1*$^{-/-}$ neonates is due to acytokinetic mitosis.

## Loss of p53 does not rescue the phenotype

To understand the role of p53 in the pathophysiology, we asked whether *p53* KO alters the phenotype of *Mcph1* KO mice. We therefore generated *Mcph1*$^{-/-}$,*Trp53*$^{-/-}$ double knockout mice (Fig. 5A). Similar altered Hb and RBC values were found in *Mcph1*$^{-/-}$ mice with or without *Trp53* (Hb 87 vs 82 g/L; RBC 1.16 vs 1.23 × 10$^{12}$/L, respectively) (Fig. 5B,C). These results indicate that *Trp53* inactivation did not correct the impairment of erythroid terminal differentiation in *Mcph1*$^{-/-}$ neonates. On the contrary, cell cycle analysis of erythroid progenitors from neonatal livers showed that the cytokinesis failure was exacerbated in *Mcph1*$^{-/-}$,*Trp53*$^{-/-}$ compared to *Mcph1*$^{-/-}$,*Trp53*$^{+/+}$ mice with an increased production of hyperploid cells (22.6% vs 12.5%, respectively), while a decrease in sub-G1 cells (DNA<2n) was observed (2.3% vs 6.0%, respectively) (Fig. 5D,E). These results show that the *Trp53* activation observed in *Mcph1*$^{-/-}$ erythroid progenitor is not the cause of the impaired erythroid terminal differentiation but rather a consequence, with *Trp53* probably exerting a protective role on *Mcph1* KO-induced polyploidization.

## Defects identified in erythroid progenitors are also found in neural progenitors

*Mcph1*$^{-/-}$ mice display microcephaly with a reduction in brain thickness of about 40% in KO mice compared to WT (110.2 μm vs 63.2 μm, *p* < 0.0001) (Fig. 6A,B). We next wondered whether the pathogenic mechanisms evidenced in erythroid progenitors also apply to neural progenitors. The neocortex size is determined by neuronal output, which is governed by the proliferative capacity of NPCs. *Mcph1* expression is prominent during the early stages of neocortical development and decline from E10.5 to E13.5, with no expression detected at E14.5 in the embryonic brain (Journiac et al, 2020). Accordingly, we performed our experiments on NPCs obtained after culturing of dorsal telencephalon samples from E12.5 embryos. Cytomorphological examination of the primary culture from *Mcph1*$^{-/-}$ brain showed frequent binucleated cells or enlarged nuclei (Fig. 6C,D). These alterations were very similar to those observed in erythroid precursors and were consistent with a cell division defect. The cell division defect was further highlighted by double immunofluorescence staining for Tpx2, a protein that interacts with spindle microtubules, and Aurora kinase B (AurkB),

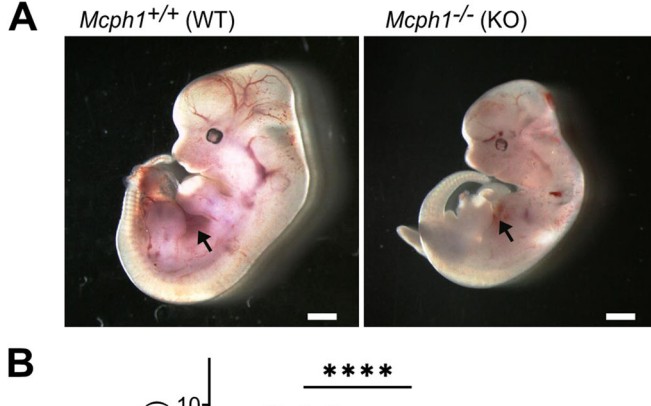

**A** Mcph1^+/+ (WT)  Mcph1^-/- (KO)

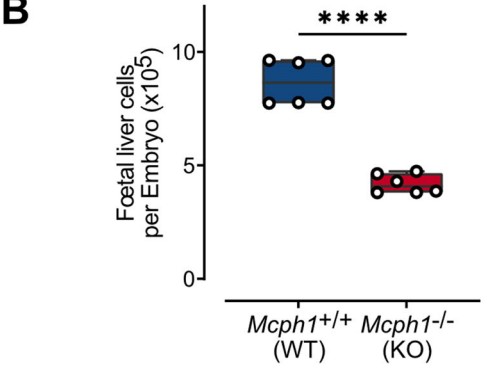

**B**

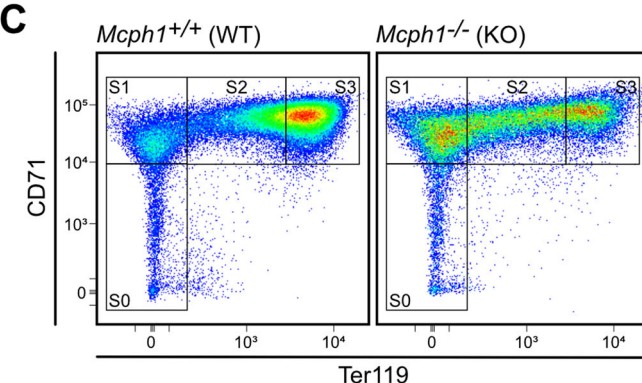

**C** Mcph1^+/+ (WT)  Mcph1^-/- (KO)

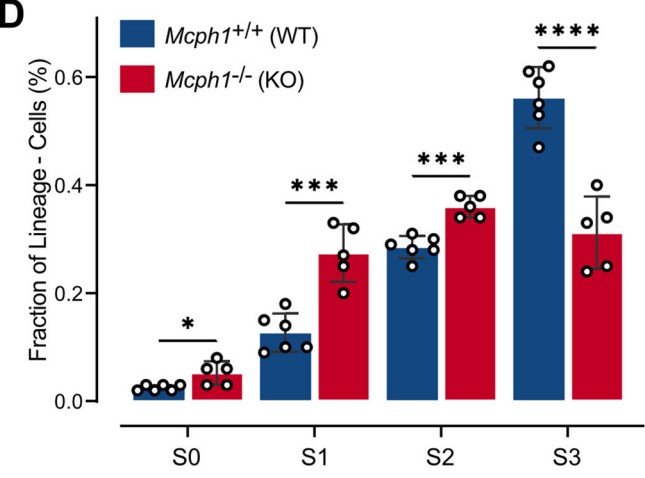

**D**

**Figure 2. Defects in terminal erythroid differentiation arise during fetal development.**

Wild type (WT, Mcph1^+/+) and knockout (KO, Mcph1^-/-) mouse fetuses from multiple litters were collected after 12.5 days of development (E12.5). (A) Photograph of Mcph1^+/+ (left) and Mcph1^-/- (right) mouse fetuses. Black arrows indicate fetal livers. Scale bar = 10 mm. (B) The cellularity of fetal livers from Mcph1^+/+ ($n = 6$) and Mcph1^-/- ($n = 6$) E12.5 animals was determined. Median, 25th, 75th percentiles, min and max. Two tailed t-test ($t = 10.04$, df = 10), **** <0.0001. (C, D) CD71 and Ter119 expression was studied by flow cytometry on Lin^- cells isolated from fetal livers from Mcph1^+/+ ($n = 6$) and Mcph1^-/- ($n = 5$) E12.5 animals. The proportion of S0, S1, S2, S3 subpopulations was determined. Mean ± SD of 3 experiments. Multiple t-test for p-values (df = 9). * <0.05, *** <0.001, **** <0.0001. Source data are available online for this figure.

a cell cycle kinase that localizes to the midbody prior to abscission, the final stage of cytokinesis, which showed frequent disorganized mitotic spindles at the ventricular border in brain sections of Mcph1 KO embryos (Fig. EV4A). Immunofluorescent labeling of the p21 protein showed a higher number of p21-expressing NPC in Mcph1^-/- primary cultured as compared with Mcph1^+/+. Of note, p21 was mostly expressed in cells with enlarged nuclei or binucleated cells. In vivo immunofluorescence on fetal brain sections at E12.5 showed a p21 protein expression in endothelial cells (CD31+) in both WT and KO embryos. Interestingly, p21 protein labeling of non-endothelial cells (CD31−) was exclusively observed in the cerebral cortex of Mcph1^-/- embryos (Fig. 6E). Furthermore, no signs of senescence or DNA damage were observed, as evidenced by the absence of p16 staining or γH2AX foci, in the fetal KO brain (Fig. EV4B,C).

Gene expression profiling of NPC from Mcph1 KO and WT fetuses showed a remarkable enrichment of p53 target genes expression in neural cells from KO mice (NES = 1.58, $p = 0.0001$, FDR = 0.007) (Fig. 6F). Moreover, differential analysis revealed that 5 out of the 10 p53 targets overexpressed in Mcph1 KO erythroid precursors were also overexpressed in the Mcph1 KO neuroprogenitors ($p < 0.01$, FDR < 0.3) (Fig. 6G). RT-ddPCR analysis of a larger number of samples ($n = 8$) confirmed a 3.7-fold Cdkn1a overexpression at the RNA level in neuroprogenitor cells from KO mice ($p = 0.0005$) (Fig. 6H).

We next investigated whether the neural phenotypes observed in KO mice in NPC are also independent of p53. Brain sections from newborn mice showed that brain thickness in Mcph1^-/-,Trp53^-/- mice is reduced to the same extent as in Mcph1^-/-,Trp53^+/+ compared to Mcph1^+/+,Trp53^+/+ mice, ruling out that p53 inactivation is able to rescue the microcephaly (Fig. 7A). Importantly, no p21 expression was detected in Trp53 knockout NPCs, suggesting that p21 expression is indeed a consequence of increased activation of the p53 pathway (Fig. EV5). NPCs cultured from Mcph1^-/-,Trp53^-/- exhibited giant nuclei (Fig. 7B,C) indicating that the absence of p53 promotes polyploidization in these cells, as observed in erythroid progenitors.

Overall, alterations observed in NPC are highly reminiscent of those observed in erythroid progenitors, supporting that a similar pathophysiological mechanism underpins the erythroid and neurological defect in Mcph1 KO mice.

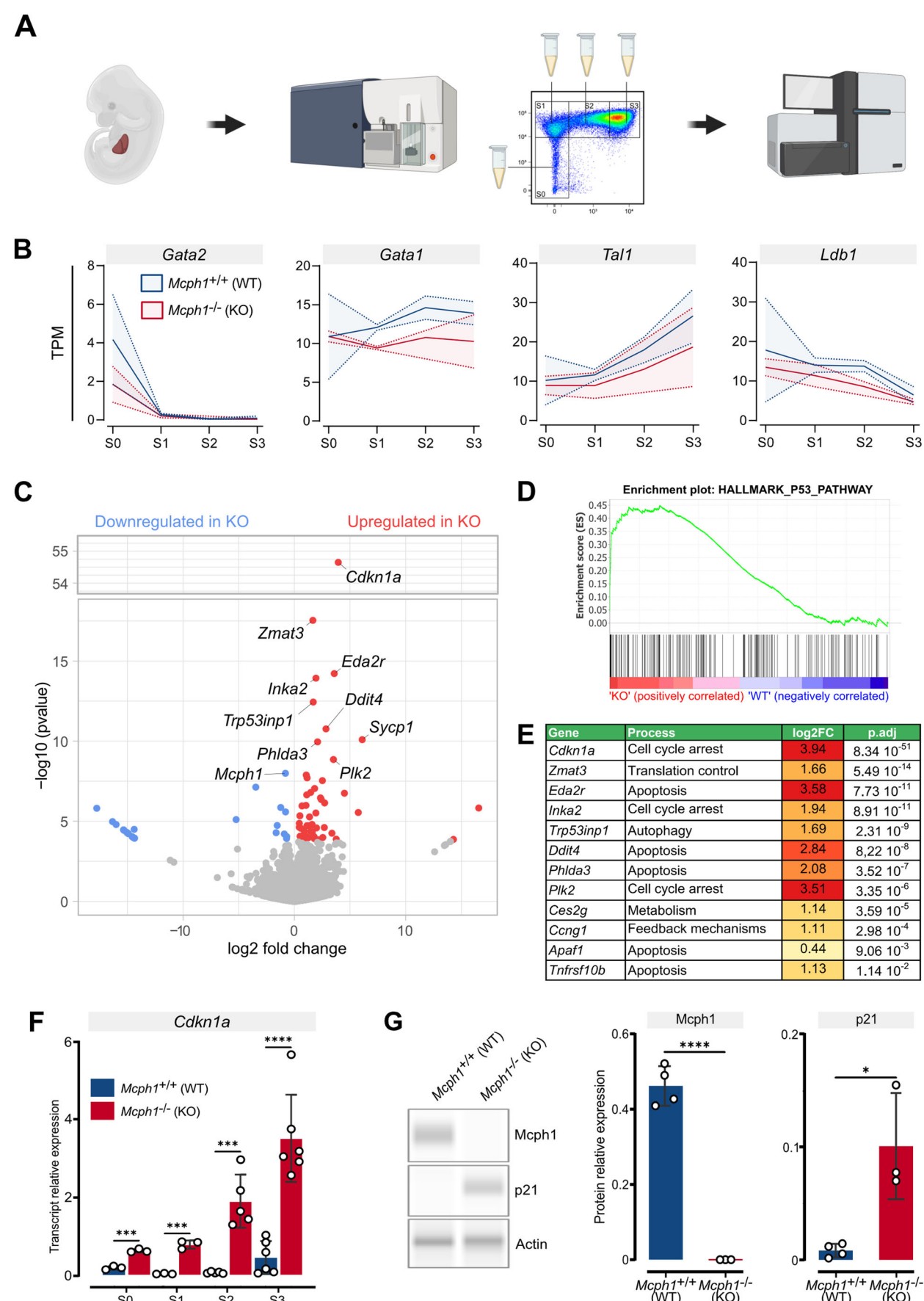

◄ **Figure 3. Loss of Mcph1 leads to overexpression of the p53 target genes.**

RNA extracted from S0, S1, S2, and S3 subpopulations sorted from fetal liver of wild type (WT, *Mcph1$^{+/+}$*) and knockout (KO, *Mcph1$^{-/-}$*) mice (E12.5) were sequenced. (A) Schematic representation of the experiment. Created with BioRender.com. (B) Expression data of major genes involved in erythroid differentiation (*Gata2, Gata1, Tal1, Ldb1*) in cells isolated from fetal liver of WT (blue) and KO (red) mice. Data expressed in transcript per million (TPM). Mean ± SD (filled area) of 2 experiments. Enrichment and differential expression analysis was performed using RNA-seq data. (C) Volcano plot of log10 (*p* value) versus log2 fold change obtained from differential expression analysis performed with DESeq2. Colored dots represent p-adjust value < 0.05, genes downregulated in KO (blue) and genes upregulated in KO (red) and not significantly dysregulated (gray). The 10 most dysregulated genes were labeled. (D) GSEA analysis on hallmark genes set comparing KO versus WT. Enrichment plot of the p53 pathway. (E) Focus on the list of p53 census genes identified in the differential expression analysis performed with DESeq2. 10 genes over 343 were significantly overexpressed (p.adj<0.05). Genes are ordered by p-adust value. The color of the Log2fold change (Log2FC) from light yellow to bright red indicates the level of overexpression in KO compared to WT. (F) *Cdkn1a* transcript was quantified by ddRT-PCR in all erythroid progenitors from S0 to S3. The data represented on the histogram show relative quantification of *Cdkn1a* transcript to *Tbp* transcript. *Mcph1$^{+/+}$*: S0 ($n = 3$), S1 ($n = 3$), S2 ($n = 5$), S3 ($n = 6$). *Mcph1$^{-/-}$*: S0 ($n = 3$), S1 ($n = 3$), S2 ($n = 5$), S3 ($n = 6$). Mean ± SD. t-test for *p*-value. S0: $t = 11.83$, df = 4; S1: $t = 12.46$, df = 4; S2: $t = 6.02$, df = 8; S3: $t = 6.27$, df = 10, *** <0.001, **** <0.0001. (G) Protein quantification of p21 encoded by *Cdkn1a* was assessed on protein extract obtain from total fetal liver (E12.5). Picture shows western-blot-like image of representative samples from WT and KO mice. Histogram compiling the relative areas under the curve (AUC) of Mcph1/actin and p21/actin ratios obtained from *Mcph1$^{+/+}$* ($n = 4$) and *Mcph1$^{-/-}$* ($n = 3$) animals. Mean ± SD. t-test for *p*-value. Mcph1: $t = 14.84$, df = 5; p21: $t = 4.025$, df = 5, * <0.05, **** <0.0001. Source data are available online for this figure.

## Neurogenesis and hematopoiesis share an important requirement for the cell replication machinery

Our data raise the question of the link that might exist between two processes as different as neurogenesis and erythropoiesis.

The developmental transcriptome at the single-cell level of 17 mouse tissues and organs from embryonic day 10.5 to birth was recently reported (He et al, 2020b, Data ref: He et al, 2020a). A cluster of 801 genes, which remarkably includes *Mcph1*, was identified by the authors as a temporal driver of developmental changes. Although ubiquitously expressed, this gene cluster was identified as most highly expressed across ontogeny in hematopoietic and neural tissues. In search of biological processes common to these two processes, we further analyzed this group of genes.

Enrichment analysis using Enrichr (Kuleshov et al, 2016) on KEGG pathways revealed a significant enrichment in genes involved in cell cycle, DNA replication, and repair pathways (e.g., Homologous recombination, Base excision repair, Mismatch repair) (Fig. 8A). Enrichment analysis on the Online Mendelian Inheritance in Man (OMIM) database indicated major association with genes signatures of Fanconi Anemia, microcephaly and anemia (Fig. 8B).

Taken together, these data emphasize that, despite their major functional differences, hematopoietic and neural development share during embryogenesis the strong expression of a transcriptional program of cell replication. This common dependence of neurogenesis and hematopoiesis on the proper functioning of the replicative machinery is likely to underpin the unique pathological link between microcephaly and anemia.

## Discussion

Primary Microcephaly (MCPH) is a neurodevelopmental disorder characterized by a smaller brain size at birth due to impaired proliferation and differentiation of neuroprogenitor cells during embryonic neurogenesis. To date, 30 loci have been identified as being causal for MCPH. However, beyond this genetic heterogeneity, there is a striking functional consistency, with most of the mutations responsible for microcephaly having the common consequence of compromising cell division by disrupting mitotic structures such as the centrosome or the mitotic

spindle (Phan and Holland, 2021). Another common theme is that microcephaly is frequently reported in patients with iBMF such as Fanconi anemia or dyskeratosis congenita. This recurrent association is not fully understood and little is known about the role of microcephaly genes in the hematopoietic development.

Questioning the possible role of MCPH1 in hematopoietic development, we found that in addition to its well-known involvement in microcephaly, loss of *Mcph1* leads to macrocytic dyserythropoietic anemia resulting from acytokinetic mitosis during terminal erythroid differentiation, thus preventing the production of sufficient numbers of mature red blood cells.

Mcph1 is known to have various functions related to the cell cycle, including chromosome segregation, DNA repair, and DNA recombination (Venkatesh and Suresh, 2014; Liang et al, 2010). Our findings on the mechanisms of the anemia developed by *Mcph1* KO mice provide additional insight into the function of MCPH1 in fetal development, suggesting a critical role in the regulation of cytokinesis, a step required for cell proliferation. Most interestingly, polyploidization resulting from acytokinetic mitosis is also found in NPC, highlighting that a common pathophysiological mechanism explains the abnormalities in neural and erythroid progenitors.

Cytokinesis defects can result from chromosome segregation errors (e.g., merotelic attachment, multipolar spindles, dicentric chromosome, DNA bridges), or altered expression of cytokinesis drivers or regulators (Lens and Medema, 2019). Notably, Cdk1 plays a crucial role in synchronizing anaphase and cytokinesis (Lau et al, 2021b). Mcph1 has been found to exert a positive regulation on the Chk1-Cdc25a-Cdk1 signaling pathway through the modulation of Chk1 expression (Xu et al, 2004; Yang et al, 2008; Passemard et al, 2011; Lin et al, 2005) and Cdc25a stability (Alderton et al, 2006) which may explain the role of Mcph1 in cytokinesis. The report in *Chk1*-haploinsufficient mice of an hematopoietic disorder highly reminiscent of the one we evidenced in our *Mcph1*-KO mice, with multiple mitotic defects, increased binucleation, and frequent macrocytic anemia during their first year of life (Peddibhotla et al, 2009), supports this hypothesis.

Cytokinesis failure, has previously been identified as a cause of congenital dyserythropoietic anemia (CDA), a bone marrow failure syndrome restricted to the erythroid lineage in humans (Traxler and Weiss, 2013; Liljeholm et al, 2013). In mice, inactivation of

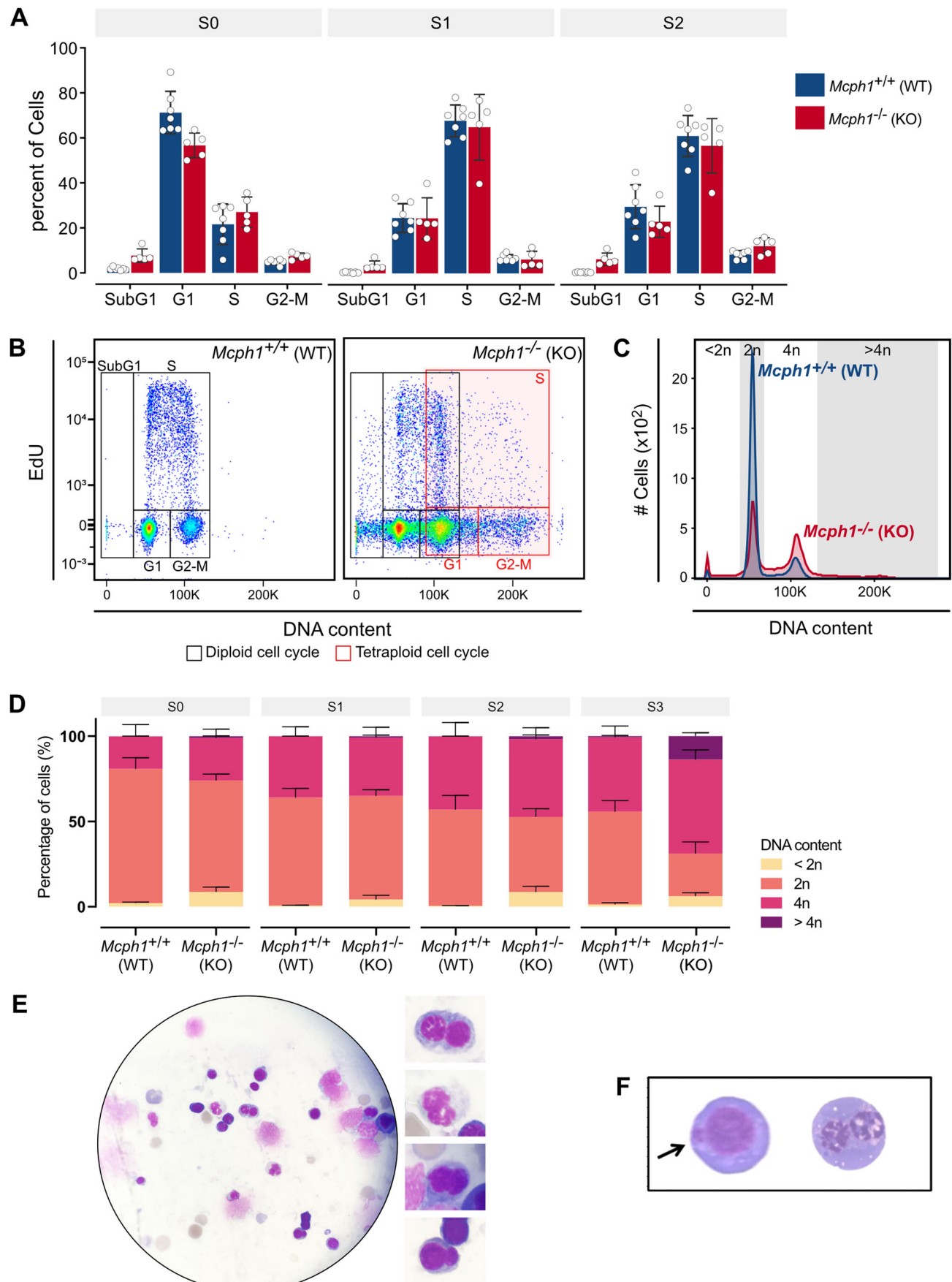

**Figure 4. Complete inactivation of *Mcph1* results in a cell cycle defect in S3 erythroid progenitors.**

Liver were collected from neonates of several litters to perform cell cycle analysis. (A) EdU incorporation and FxCycle labeling analysis by flow cytometry on the subset of erythroid progenitors defined by Ter119 and CD71 expression in Lin- cells allow determination of the percentage of cells in each cycle phase. Data are compiled as an histogram for each population S0, S1, and S2. Mean ± SD of 3 experiments. (B) Illustration of EdU incorporation and DNA content in S3 subset of erythroid progenitors from representative samples isolated from wild type (WT, *Mcph1*$^{+/+}$) and knockout (KO, *Mcph1*$^{-/-}$) animals. (C) DNA content (<2n, 2n, 4n, >4n) was estimated based on FxCycle incorporation by flow cytometry in the S3 subset. Graph represents number of cells and DNA content in the S3 subset in representative samples from WT (blue) and KO (red) mice. (D) Stack histograms compile percent of cells with <2n, 2n, 4n and >4n DNA in S0, S1, and S3 subset isolated from wild type (WT, *Mcph1*$^{+/+}$) ($n = 8$) and knockout (KO, *Mcph1*$^{-/-}$) ($n = 6$) animals. Mean ± SD of 3 experiments. May-Grünwald-Giemsa-stained erythroid cells (E) or S3 erythroid progenitor cells with >4n DNA content (F) sorted by FACS from knockout (KO, *Mcph1*$^{-/-}$) sample. Black arrow points to a micronucleus. Source data are available online for this figure.

*RhoA* or *mDia2*, essential component for cytokinesis, recapitulate this phenotype (Konstantinidis et al, 2015; Watanabe et al, 2013). It is unclear why a defect in cytokinesis primarily affects erythropoiesis, but such observations highlight the importance of this process in erythroid differentiation. Similarly, the regulation of cytokinesis appears to be essential for neural development. For instance, inactivation of *mDia2* in mouse cerebral cortex profoundly disrupts neurogenesis, depleting cortical progenitors and neuron (Lau et al, 2021a). The discovery of Mcph1 involvement in cytokinesis suggests that abnormalities in cytokinesis may play a more prominent role in primary microcephaly than previously thought. In this regard, it is interesting to note that although not brought to the forefront by the authors, there is other evidence suggestive of the involvement of altered cytokinesis in microcephaly. For instance, loss of function mutations in *Citk* (MCPH17), the kinase downstream of RhoA, cause apoptosis due to DNA damage but also cytokinesis failure in neural progenitors (Bianchi et al, 2017). Likewise, higher numbers of binucleated cells were observed in patients with *KIF14* deficiency (MCPH20) (Moawia et al, 2017).

Our findings pointed to the p53 pathway as a potential player in the *Mcph1*-driven defect, both in erythroid and neural progenitors. The activation of p53 is a constant in both microcephaly and iBMF. However, the role of p53 in these disorders is complex. In several setting, inactivation of p53 has allowed to rescue the phenotype. Furthermore, p53 germline hyperactivation has recently be shown to cause a syndrome in humans that associates anemia and microcephaly (Kumar et al, 2022), reinforcing the idea that p53 activation alone can be the driver of both anemia and microcephaly. However, in our model, p53 inactivation could not restore either the hematological or the neural phenotype. Instead, it exacerbated polyploidization by increasing the number of acytokinetic mitoses in erythroid precursors and NPCs. This is consistent with Tátrai's study showing that loss of *Cdk5rap2* (MCPH3), another MCPH-associated gene, induces the formation of tetraploid erythroblats. Although p53 is activated in this model as well, it is not responsible for the macrocytic anemia that is observed. (Tátrai and Gergely, 2022). Overall, disorders can be separated into two main groups depending on whether the phenotype is reversed by p53 inhibition or not. For instance, phenotypes caused by mutations in the iBMF genes *Fancd2* and *Rps14* (Ceccaldi et al, 2012; Barlow et al, 2010) or in the microcephaly-related genes *Aspm, Cep63*, and *Cenpj* (Williams et al, 2015; Phan et al, 2021; Insolera et al, 2014) are rescued by the inactivation of p53 whereas congenital anemia and microcephaly caused by the inactivation of *Cdk5rap2* (Tátrai and Gergely, 2022) or *Cep135* (MCPH8) (González-Martínez et al, 2022) genes are not. It is known that p53-

response leading to either survival or death depends on the type of damage experienced by the cell, its severity, and duration in time (Rizzotto et al, 2021). In this regard, it is noteworthy that unlike other MCPH genes *Cdk5rap2, Cep135,* and *Mcph1* loss of function induce an increase in centrosome numbers (Megraw et al, 2011). Supernumerary centrosomes can occur via several mechanisms, including deregulated centrosome duplication, uncontrolled splitting of centriole pairs and cytokinesis failure (Nigg, 2002). In this later case, it induces a p53 stabilization that triggers G1 arrest through PIDDosome activation instead of cell death (Fava et al, 2017). Therefore, we can assume that in *Mcph1*-deficient cells, the reduced cell expansion is the result of a decrease in the number of cells produced, with 1 mother cell giving 1 daughter cell instead of 2 due to cytokinesis failure, rather than their apoptosis, as prominent in iBMF and microcephaly. These results underscore the importance of a thorough evaluation of the pathophysiological role of the p53 response in each type of microcephaly and iBMF, which is of major importance for future therapeutic strategies.

In conclusion, we provide here a striking example of how alteration of a single gene affects the development of both hematopoietic and neurological tissues, and show the importance of the critical role of proper mitosis for efficient neurogenesis and erythropoiesis. This also highlight the existence of biological mechanisms whose importance is electively shared by these two processes. In this respect, our further analysis of the interesting data of He et al provides a broader perspective. Indeed, we have identified *Mcph1* as part of a group of genes that, although ubiquitous, are particularly intensely expressed in developmental neurogenesis and hematopoiesis compared to all other tissues. This shared transcriptional program is related to the cell division machinery but also to anemia and microcephaly. This observation should be considered in light of the common challenge that these two tissues face despite all their functional differences. Both neural and erythroid lineages must rapidly produce a very large number of cells during fetal development. Indeed, neurons are generated at staggering rates of approximately 3.86 million each hour over the course of developmental neurogenesis (Silbereis et al, 2016). Similarly, it has been estimated that red blood cell mass increases 70-fold in fetal mice between embryonic day 12.5 (E12.5) and E16.5 of gestation (Russel et al, 1968). It is therefore clear that any disruption of the cell cycle will have an elective impact on the efficiency of both neurogenesis and erythropoiesis. These common massive proliferative requirements are likely to explain the recurrent association of disorders affecting both systems when a genetic alteration impairs the cell cycle machinery.

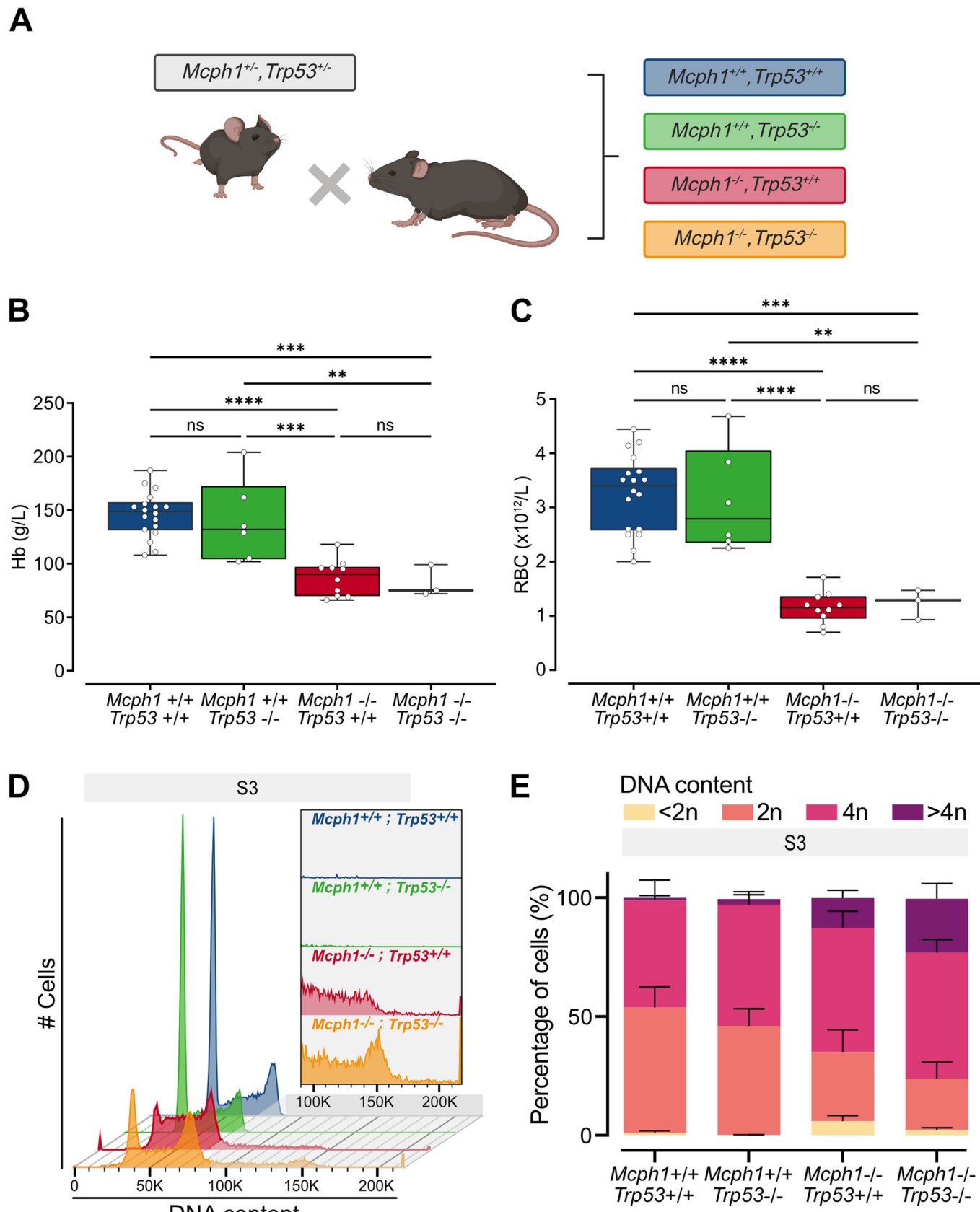

◄ **Figure 5. Inactivation of p53 does not correct the erythropoiesis defects in *Mcph1*$^{-/-}$ mice.**

*Mcph1*$^{+/-}$ and *Trp53*$^{+/-}$ mice were crossed to generate double heterozygous animals. (A) The *Mcph1*$^{+/+}$,*Trp53*$^{+/+}$ ; *Mcph1*$^{+/+}$,*Trp53*$^{-/-}$ ; *Mcph1*$^{-/-}$,*Trp53*$^{+/+}$ and *Mcph1*$^{-/-}$, *Trp53*$^{-/-}$ animals resulting from the mating of the double heterozygotes were analyzed. Schematic representation of the experiment. Created with BioRender.com. (B, C) Peripheral blood were collected from *Mcph1*$^{+/+}$,*Trp53*$^{+/+}$ ($n = 18$), *Mcph1*$^{+/+}$,*Trp53*$^{-/-}$ ($n = 6$), *Mcph1*$^{-/-}$,*Trp53*$^{+/+}$ ($n = 10$) and *Mcph1*$^{-/-}$,*Trp53*$^{-/-}$ ($n = 3$) neonates of several litters to perform complete blood count. Boxes and whiskers represent hemoglobinemia (Hb) in g/L and red blood cell (RBC) counts in ×10$^{12}$/L. Median, 25th, 75th percentiles, min and max. One-way anova, Hb: $F_{(3,33)} = 17.7$, RBC: $F_{(3,33)} = 27.3$, ** <0.01, *** <0.001, **** <0.0001. (D) DNA content (<2n, 2n, 4n, >4n) was estimated on the basis of FxCycle incorporation by flow cytometry in the S3 subset in representative samples from *Mcph1*$^{+/+}$,*Trp53*$^{+/+}$ (Blue), *Mcph1*$^{-/-}$,*Trp53*$^{-/-}$ (Green), *Mcph1*$^{-/-}$,*Trp53*$^{+/+}$ (Red) and *Mcph1*$^{-/-}$,*Trp53*$^{-/-}$ (Yellow). (E) Histograms compile percent of cells with <2n, 2n, 4n and >4n DNA in S3 subset isolated from *Mcph1*$^{+/+}$,*Trp53*$^{+/+}$ ($n = 11$), *Mcph1*$^{+/+}$,*Trp53*$^{-/-}$ ($n = 3$), *Mcph1*$^{-/-}$,*Trp53*$^{+/+}$ ($n = 7$) and *Mcph1*$^{-/-}$,*Trp53*$^{-/-}$ ($n = 6$). Mean ± SD of 3 experiments. Source data are available online for this figure.

# Methods

### Reagents and tools table

| Reagent/Resource | Reference or Source | Identifier or Catalog Number |
|---|---|---|
| **Experimental models** | | |
| *Mcph1*$^{tm1.2Kali}$ (*M. musculus*) | Liang et al, 2010 | |
| *Trp53*$^{tm1Tyj}$ | Jackson Laboratory | Cat #002101 |
| **Antibodies** | | |
| FITC-coupled rat anti-Gr1 | BioLegend | Cat #108406 |
| FITC-coupled rat anti-CD11b | BioLegend | Cat #101206 |
| FITC-coupled rat anti-CD8a | BioLegend | Cat #100706 |
| FITC-coupled rat anti-CD4 | BioLegend | Cat #100406 |
| FITC-coupled rat anti-B220 | BioLegend | Cat #103206 |
| APC/Cy7-coupled rat anti-CD41 | BioLegend | Cat #133928 |
| PE-coupled rat anti-CD71 | BioLegend | Cat #113808 |
| PerCP/Cy5.5 rat anti-TER-119 | BioLegend | Cat #116228 |
| FITC-coupled hamster anti-CD3ε | BioLegend | Cat #152304 |
| Rabbit anti-Mcph1 | Cell Signaling | Cat #4120 |
| Mouse anti-beta-Actin | Novus biological | Cat #NB600-501 |
| Rabbit anti-p21/Cdkn1a | Abcam | Cat #ab188224 |
| Mouse anti-AurkB | BD Transduction laboratories | Cat #611083 / RRID:AB_398396 |
| Rabbit anti-Tpx2 | Proteintech | Cat #11741-1-AP / RRID:AB_2208895 |
| Rat anti-CD31 | BD Pharmingen | Cat #550274 / RRID:AB_393571 |
| Rabbit anti-Anillin | Abcam | Cat #ab154337 |
| Mouse anti-γH2A.X (P- S139) | Abcam | Cat #ab22551 |
| Cy3-coupled Goat anti-rabbit IgG | Jackson ImmunoResearch | Cat #11-165-144 / RRID:AB_2338690 |
| Cy3-coupled Mouse anti-rabbit IgG | Jackson ImmunoResearch | Cat #115-165-146 / RRID:AB_2338690 |
| Aexa488-coupled Goat anti-rabbit IgG | ThermoFisher Scientific | Cat #A11034 / RRID:AB_2576217 |
| Aexa488-coupled Mouse anti-rabbit IgG | ThermoFisher Scientific | Cat #A11029 / RRID:AB_2534088 |
| Alexa633-coupled Goat anti-Rat IgG | ThermoFisher Scientific | Cat #A21094 / RRID:AB_2535749 |
| **Oligonucleotides and other sequence-based reagents** | | |
| Tbp – HEX RT-ddPCR assay | Bio-Rad | Cat #dMmuCPE5124759 |
| Cdkn1a – FAM RT-ddPCR assay | Bio-Rad | Cat #dMmuCPE5096492 |
| **Chemicals, enzymes, and other reagents** | | |
| RNeasy Mini kit | Qiagen | Cat #74104 |
| RNA 6000 Pico kit | Agilent | Cat #5067-1513 |

| Reagent/Resource | Reference or Source | Identifier or Catalog Number |
|---|---|---|
| SMARTer Stranded Total RNA-Seq Kit V2 | Takara | Cat #634412 |
| M-PER® lysis buffer | ThermoFisher Scientific | Cat #78503 |
| Halt™ Protease Inhibitor Cocktail | ThermoFisher Scientific | Cat #P2714 |
| PhosSTOP™ | Roche | Cat #4906845001 |
| PMSF | ThermoFisher Scientific | Cat #36978 |
| Protein 230 kit | Agilent | Cat #5067-1517 |
| Click-iT™ Plus EdU Alexa Fluor™ 594 Flow Cytometry assay Kit | ThermoFisher Scientific | Cat #C10646 |
| FxCycle™ Violet Stain | ThermoFisher Scientific | Cat #F10347 |
| Zombie violet™ Fixable Viability Kit | BioLegend | Cat #423114 |
| 12–230 kDa Separation Module | Biotechne | Cat #SM-W004 |
| Anti-Mouse Detection module | Biotechne | Cat #DM-002 |
| Anti-Rabbit Detection Module | Biotechne | Cat #DM-001 |
| **Software** | | |
| FlowJo™ v10.7.2 | BD Life Sciences | https://www.flowjo.com/ |
| R version 3.6.3 | | https://www.r-project.org/ |
| R studio v1.4 | | https://www.rstudio.com/ |
| Genome Browser | IGV | https://software.broadinstitute.org/software/igv/ |
| GraphPad Prism v9 | GraphPad Software | https://www.graphpad.com/ |
| Affinity Designer v1.9 | Affinity | https://affinity.serif.com/ |
| FlowJo™ v10.7.2 | BD Life Sciences | https://www.flowjo.com/ |
| Enrichr | | https://maayanlab.cloud/Enrichr/ |

## Mouse model

The *Mcph1* and *Trp53* mouse models were obtained from Kayi LI and Jackson laboratories, respectively, and have been previously described (Liang et al, 2010; Jacks et al, 1994). *Mcph1*$^{tm1.2Kali}$ null mice were generated by germline deletion of *Mcph1* exon 2 leading to a frameshift mutation of *Mcph1* (Liang et al, 2010). *Mcph1* transcript was still detected in knockout mice but expression was reduced by more than a half compared with wild-type mice (Appendix Fig. S1A,B). Protein analysis performed on the fetal liver of *Mcph1*$^{-/-}$ embryos confirmed the absence of Mcph1 protein

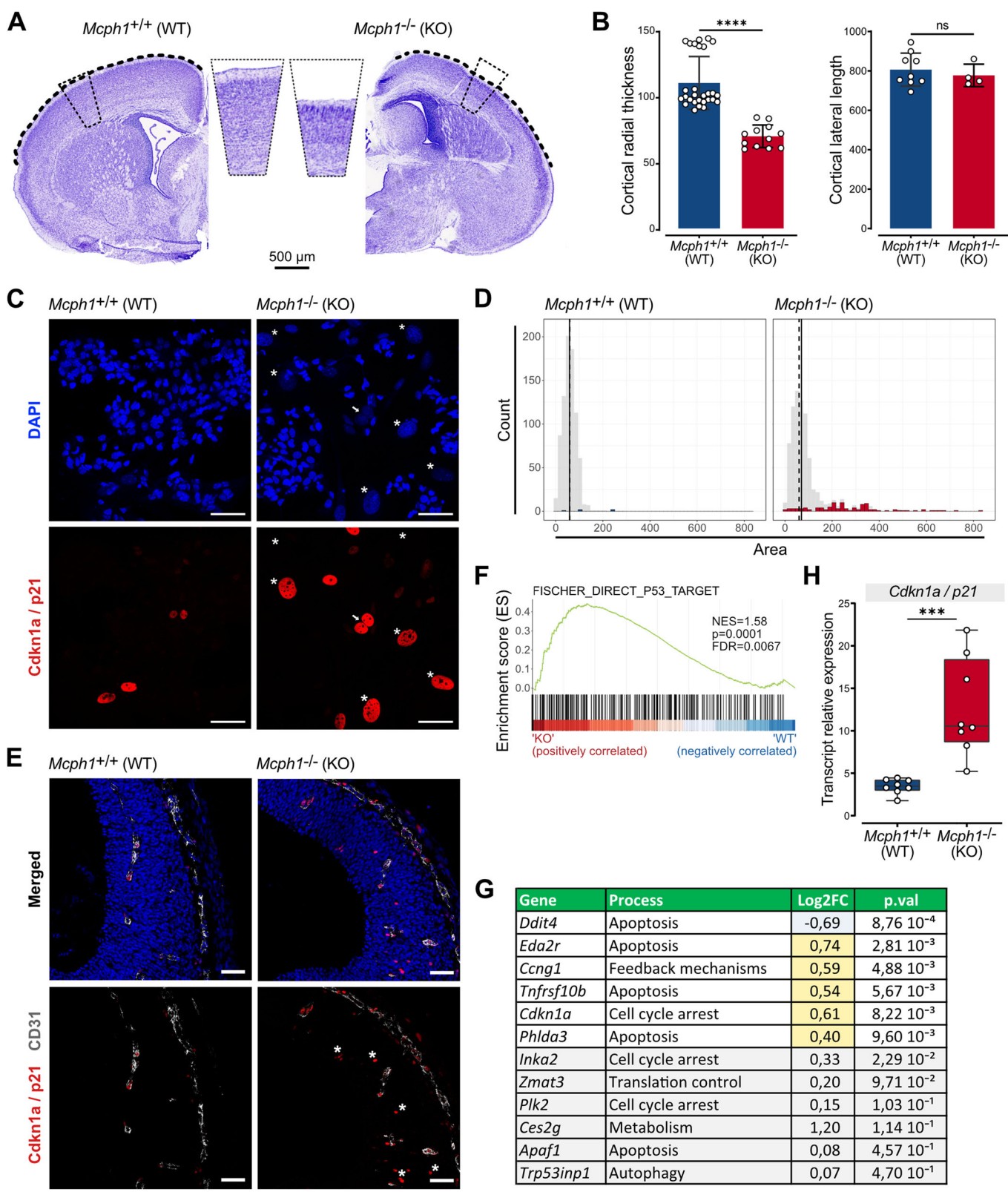

**Figure 6.   Complete inactivation of *Mcph1* results in congenital microcephaly associated with cytokinesis abnormalities and p53 activation.**

Wild type (WT, *Mcph1*$^{+/+}$) and knockout (KO, *Mcph1*$^{-/-}$) mice were examined at birth (day 0). (**A**) Histological examination of coronal slice of newborn *Mcph1*$^{+/+}$ (left) and *Mcph1*$^{-/-}$ (right) mice brains, stained with hematoxylin and eosin. The dial shows the thickness of the cerebral cortex. The dashed line represents the lateral length of the cortex. (**B**) Brains were measured. Radial thickness from Wild type (WT, *Mcph1*$^{+/+}$) ($n = 28$) and knockout (KO, *Mcph1*$^{-/-}$) ($n = 12$). Lateral length from Wild type (WT, *Mcph1*$^{+/+}$) ($n = 9$) and knockout (KO, *Mcph1*$^{-/-}$) ($n = 4$). Data were compiled in 2 histograms. Mean ± SD. t-test for *p*-values. radial: $t = 6.69$, df $= 38$; lateral: $t = 0.63$, df $= 11$, **** <0.0001. Embryonic neocortical neuroprogenitor cells (NPC) were isolated from mouse dorsal telencephalon at E12.5 gestational age. (**C**) Primary cultures of NPCs isolated from wild type (WT, *Mcph1*$^{+/+}$) and knockout (KO, *Mcph1*$^{-/-}$) mice were stained with DAPI and incubated with a p21 antibody. Asterisks (*) indicate large nuclei. The white arrow indicates a binucleated cell. Scale bar = 50 μm. (**D**) The nuclear areas were compiled in a histogram. The distribution of areas of p21-negative nuclei is shown in gray and that of p21-positive nuclei in blue for WT NPCs or in red for KO NPCs. The dotted line represents the median area and the solid line the mean area over all nuclei. (**E**) Coronal sections of *Mcph1*$^{+/+}$ (left) and *Mcph1*$^{-/-}$ (right) E12.5 fetal brains, stained with DAPI and incubated with p21 and CD31 antibodies. Asterisks (*) indicate p21 + CD31− cells. A zoom-out view of these images is shown in Fig. EV4B. Scale bar = 100 μm. RNA was extracted from primary cultures of NPCs isolated from wild type (WT, *Mcph1*$^{+/+}$) ($n = 2$) and knockout (KO, *Mcph1*$^{-/-}$) ($n = 2$). (**F**) GSEA analysis on C2: curated genes set comparing KO versus WT was performed. Enrichment plot of the Fischer direct p53 targets - meta analysis. (**G**) Focus on the 10 genes from the p53 census gene list that are overexpressed in erythroid progenitors in the differential expression analysis of NPC performed with DESeq2. (**H**) *Cdkn1a* transcript was quantified by ddRT-PCR in NPC. Boxes and whiskers represent relative quantification of *Cdkn1a* transcript to *Tbp* transcript for *Mcph1*$^{+/+}$ ($n = 8$) and *Mcph1*$^{-/-}$ ($n = 8$) mice. Median, 25th, 75th percentiles, min and max. t-test for *p*-value. $t = 4.509$, df $= 14$, *** <0.001. Source data are available online for this figure.

suggesting that this aberrant *Mcph1* mRNA is degraded by nonsense-mediated mRNA decay (Appendix Fig. S1C). *Trp53*$^{tm1Tyj}$ (Jackson laboratory #002101) were generated by germline deletion of exon 2–6 of *Trp53* leading to a frameshift mutation of *Trp53* (Jacks et al, 1994). Heterozygous *Mcph1*$^{+/-}$ mice were bred to generate wild type (*Mcph1*$^{+/+}$), heterozygous (*Mcph1*$^{+/-}$) and knockout animals (*Mcph1*$^{-/-}$) animals. *Mcph1*$^{+/-}$ mice were crossed with *Trp53*$^{+/-}$ to generate double heterozygous mice (*Mcph1*$^{+/-}$,*Trp53*$^{+/-}$). Double heterozygous mice were bred to generate wild-type (*Mcph1*$^{+/+}$,*Trp53*$^{+/+}$), simple knockout (*Mcph1*$^{-/-}$,*Trp53*$^{+/+}$, *Mcph1*$^{+/+}$,*Trp53*$^{-/-}$), and double knockout animals (*Mcph1*$^{-/-}$,*Trp53*$^{-/-}$). All mice were maintained in the C57BL/6J genetic background. Embryonic development stage was estimated by considering the day of vaginal plug formation at 0.5 days post coitum. Pregnant females were sacrificed at E12.5 for the embryo study. Newborn mice were sacrificed at day 0 to evaluate hematopoiesis at birth. Animals and embryos were genotyped by PCR on DNA extracted from hairs or tail, respectively. Blood was collected from newborns for a complete blood count on MS9-5S (MS Laboratories) and cytomorphologic examination.

All animal experimentations were reviewed and approved by the DDPP (Département De Protection des Populations) of Paris (#B75-19-01).

## Flow cytometry analysis and cell sorting

Antibodies used are listed in Reagent Table. Fetal and newborn liver cells were labeled with fluorochrome-conjugated antibodies against CD71, Ter119, and CD41 to define the erythroid progenitor subsets S0 to S5 and the megakaryocyte population as described previously (Koulnis et al, 2011). Dead cells incorporating the Zombie Violet™ Fixable Viability Kit, non-megakaryocytic and non-erythroid mature cells expressing B220, CD4, CD8a, CD11b, and/or Gr-1 (lineage cocktail, Lin) were excluded. Fluorescence-activated cell sorting (FACS) experiments were performed using LSRFortessa X-20 (BD Biosciences). Data were analyzed using FlowJo Software (BD Biosciences). The cell cycle analysis was performed by flow cytometry using a combination of Click-iT Plus EdU Alexa fluor 594 and FxCycle Violet (ThermoFisher Scientific). Cells were incubated with 10 μM of EdU for 2 h and then stained with FxCycle Violet and the cocktail of antibody described above. The same

combination of antibodies was used for cell sorting of erythroid subsets on BD FACSAria II (BD Biosciences).

## Embryonic neocortical neuroprogenitor cell primary culture

Mouse dorsal telencephalon was isolated from E12.5 brains from *Mcph1*$^{tm1.2Kali}$ WT and KO mice and processed as described by our group to obtain primary cultures of embryonic neocortical neuroprogenitor cells (Journiac et al, 2020).

## RNA extraction and purification

Total RNA was isolated from either sorted erythroid progenitor fractions from E12.5 fetal livers of both wild-type and knockout mice from the same litter using the RNeasy mini kit (QIAGEN), or from cultured neuroprogenitor cells isolated from E12.5 fetal brains using the Nucleospin RNA Plus (Macherey-Nagel). RNA quality and concentration were determined using Agilent RNA 6000 Pico assay on a 2100 Bioanalyzer (Agilent). Only RNA samples with an RNA Integrity Number (RIN) above 7.5 were used.

## RNA-seq and bioinformatics analysis

### Erythroid progenitors

cDNA Libraries were prepared using the SMARTer Stranded Total RNA-Seq Kit V2 – Pico Input Mammalian preparation kit (Takara) and then sequenced on a HiSeq2500 system (Illumina) (IntegraGen Platform). Sequencing produced ~62 million 75 bp paired-reads per sample. Quality of reads was assessed for each sample using FastQC and low-quality samples (less than 10 M reads) were removed from the final analyzed data set. The reads were mapped to the reference Mouse genome Gencode release M21 (GRCm38.p6) with STAR (Dobin et al, 2013). The transcripts expression was quantified with Salmon (Patro et al, 2017) and tximport (Soneson et al, 2015) on R Studio. Differential expression analysis was performed using DESeq2 (Love et al, 2014a). A linear model was fitted with all groups (cell stage, sex, and genotype) and contrasts carried out within each cell stage compared each mutant with the control samples. GSEA analysis was performed from DESeq2 matrix with GSEA software (Subramanian et al, 2005; Mootha et al, 2003) against Hallmark gene sets (MSigDB database v7.4) (Liberzon et al, 2015).

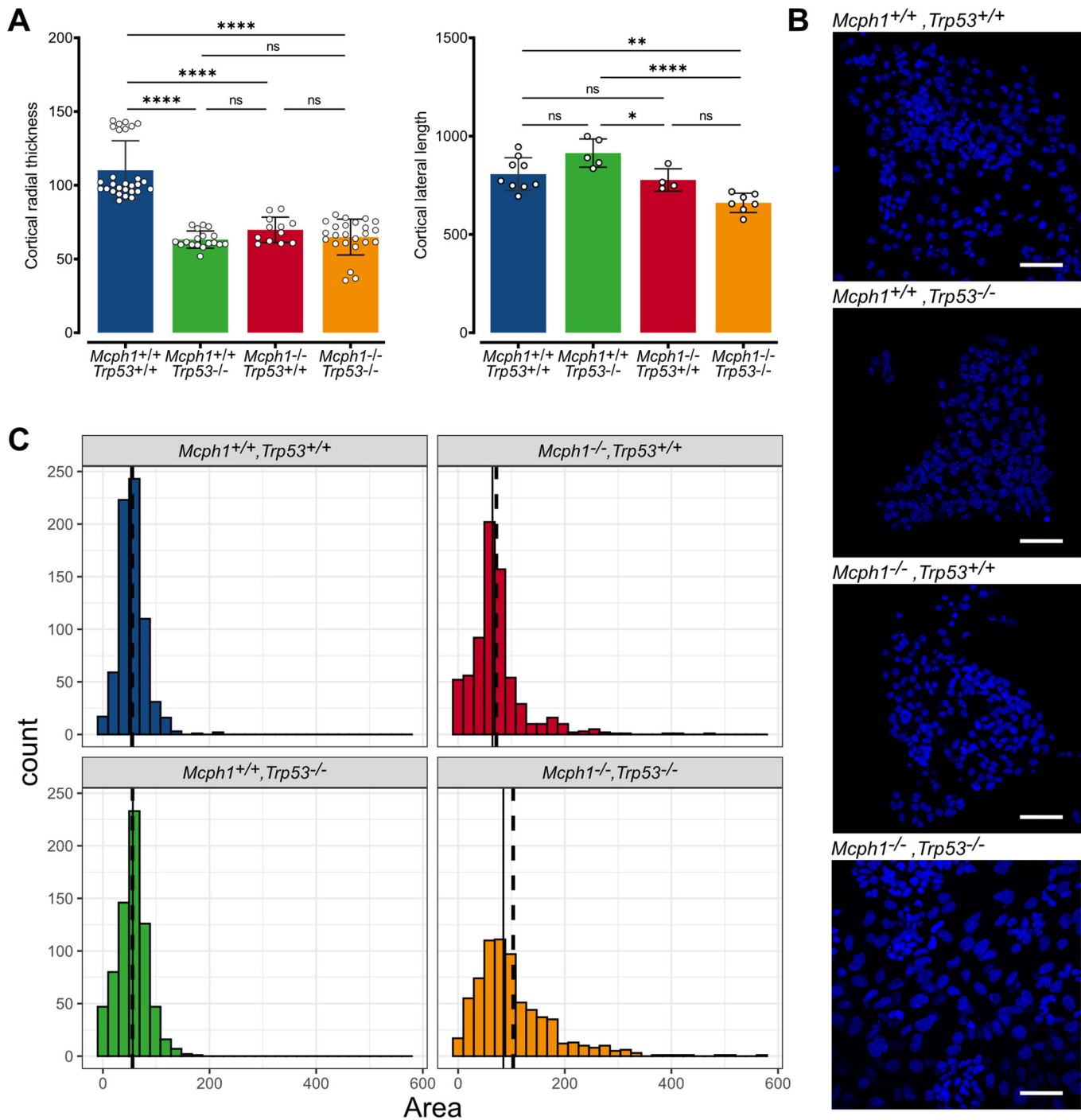

**Figure 7. Inactivation of p53 does not correct the neurogenesis defects in *Mcph1*⁻/⁻ mice.**

*Mcph1*⁺/⁺,*Trp53*⁺/⁺ ; *Mcph1*⁺/⁺,*Trp53*⁻/⁻ ; *Mcph1*⁻/⁻,*Trp53*⁺/⁺ and *Mcph1*⁻/⁻,*Trp53*⁻/⁻ mice were examined at birth (day 0). (A) Brains were measured. Radial thickness from *Mcph1*⁺/⁺,*Trp53*⁺/⁺ (n = 28); *Mcph1*⁺/⁺,*Trp53*⁻/⁻ (n = 18); *Mcph1*⁻/⁻,*Trp53*⁺/⁺ (n = 12) and *Mcph1*⁻/⁻,*Trp53*⁻/⁻ (n = 24). Lateral length from *Mcph1*⁺/⁺,*Trp53*⁺/⁺ (n = 9); *Mcph1*⁺/⁺,*Trp53*⁻/⁻ (n = 5); *Mcph1*⁻/⁻,*Trp53*⁺/⁺ (n = 4) and *Mcph1*⁻/⁻,*Trp53*⁻/⁻ (n = 7). Data were compiled in 2 histograms. Mean ± SD. One-way anova, radial: $F(3,78) = 62.12$, lateral: $F(3,21) = 13.6$, * <0.05, ** <0.01 and **** <0.0001. Embryonic neocortical neuroprogenitor cells (NPC) were isolated from mouse dorsal telencephalon at E12.5 gestational age. (B) Primary cultures of NPCs mice were stained with DAPI. Scale bar = 50 μm. (C) The nuclear areas were measured and compiled in a histogram. The distribution of areas of nuclei from *Mcph1*⁺/⁺,*Trp53*⁺/⁺ (Blue), *Mcph1*⁺/⁺,*Trp53*⁻/⁻ (Green), *Mcph1*⁻/⁻,*Trp53*⁺/⁺ (Red) and *Mcph1*⁻/⁻,*Trp53*⁻/⁻ (Yellow) NPCs were plotted. The dotted line represents the median area and the solid line the mean area over all nuclei. Source data are available online for this figure.

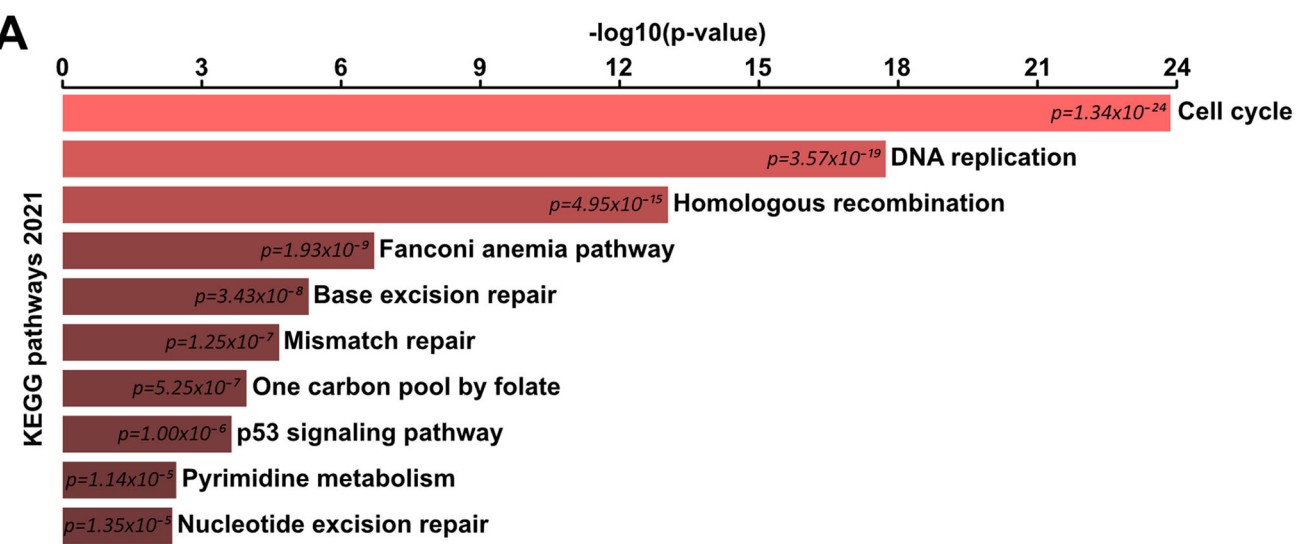

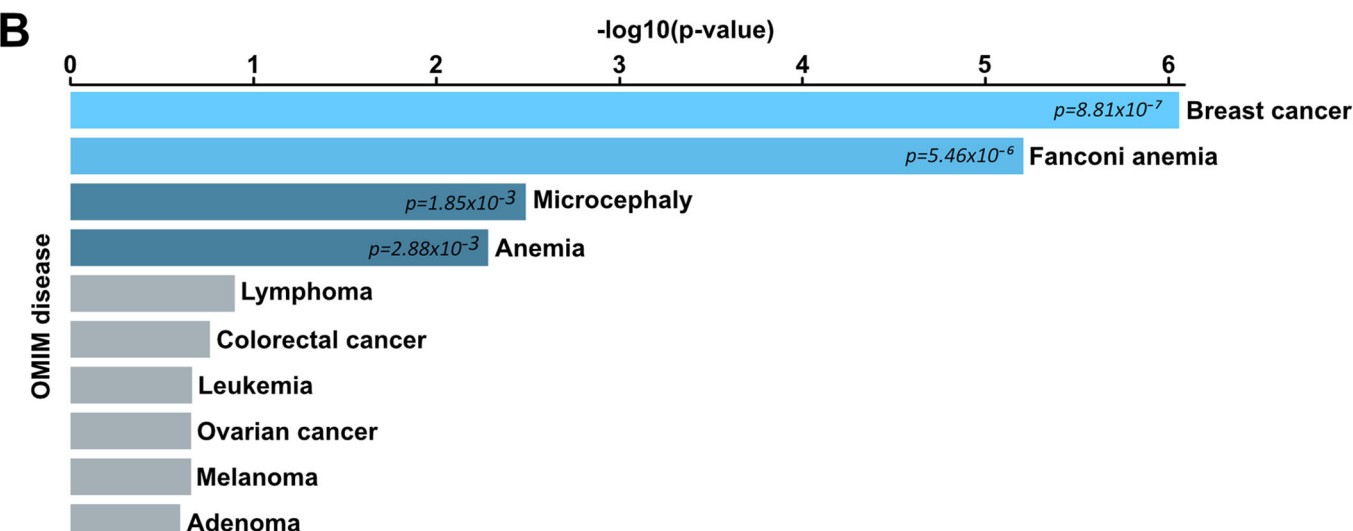

**Figure 8. Optimal cell division is a common requirement for normal neurogenesis and hematopoiesis.**

Gene set identified as a temporal driver during neurogenesis and hematopoiesis by He et al, 2020b (Data ref: He et al, 2020a) was used for enrichment analysis using Enrichr (A) KEGG pathways revealed that this set of genes and (B) Online Mendelian Inheritance in Man (OMIM).

*Neuroprogenitors*

cDNA Libraries were prepared with NEBNext Ultra II Directional RNA Library Prep Kit for Illumina protocol according supplier recommendations. Sequencing was then carried out in Paired End 100 bp reads on an Illumina NovaSeq sequencer. Quality of reads was assessed for each sample using FastQC and low-quality samples (less than 10 M reads) were removed from the final analyzed data set. The reads were mapped to the reference Mouse genome Gencode release Gencode vM24 annotation with STAR (Dobin et al, 2013). The Bioconductor edgeR package was used to import raw counts into R statistical software. Differential expression analysis was performed using the Bioconductor limma package and the voom transformation. Gene list from the differential analysis was ordered by decreasing log2 fold change. Gene set enrichment analysis was performed by clusterProfiler::GSEA function using the fgsea algorithm.

**RT-ddPCR**

RNA transcript quantification was performed by one-step RT-ddPCR on the QX200 Droplet Digital PCR System (Bio-Rad). Assays were listed in Reagent Table. Mouse *Tbp* expression was used as internal control.

**Protein analysis**

Isolated liver cells were lysed using M-PER® lysis buffer supplemented with a protease inhibitor cocktail (ThermoFisher Scientific), PhosSTOP™ (Roche), and PMSF (ThermoFisher Scientific) or RIPA lysis buffer. Total protein was quantified using the Protein 230 protein electrophoresis kit (Agilent) on the Bioanalyzer 2100 instrument (Agilent). Proteins were analyzed using the Wes

Simple Western Analyzer (Biotechne) using antibodies listed in Reagent Table.

## Immunofluorescence

NPC grown on cover slide were fixed in 4% PFA, rinsed in PBS, permeabilized in PBT during 15 min, and saturated with PBS containing 10% of normal goat serum (PBG) for 1 h. Primary antibody targeting p21 was diluted in PBG, incubated overnight, and rinsed in PBS. Secondary antibodies diluted in PBG containing 1 µg/ml DAPI (4',6-diamidino-2-phénylindole) were applied for 30 min at room temperature, washed in PBS, and coverslips were mounted in Fluoromount-G (SouthernBiotech). Imaging was performed using a Leica TCS SP8 confocal scanning system (Leica Microsystems) equipped with 405-nm Diode, 561-nm DPSS lasers. Eight-bit digital images were collected sequentially from a single optical plane using a Leica HC PL APO 20x, 40x or HC PL APO CS2 63x oil immersion lens (numerical aperture of 1.40) for cell analysis. Pictures with composite colors and the corresponding Tiff files were generated on ImageJ. DAPI and p21-positive nuclei count and area measurement was performed using the StarDist plugin on ImageJ/Fiji (Schmidt et al, 2018).

Whole E12.5 embryos and livers collected on 13.5 embryos were fixed in 4% paraformaldehyde (PFA, Sigma-Aldrich). Tissues were cryo-protected in 30% sucrose in phosphate-buffered saline (PBS), embedded in gelatin before freezing and kept at −80 °C until use. Serial 6- to 16-micron-thick sections were cut on a Leica CM 3050 cryostat.

## Statistical analysis

All experiments were replicated at least three times. Statistical analyses were performed with GraphPad Prism software. Data are shown as mean ± standard deviation (SD). A 2-tailed unpaired Student t test or multiple t-test were used to compare the mean of 2 groups, and a 2-way analysis of variance (ANOVA) was used for multiple comparisons. $p < 0.05$ was considered statistically significant.

# Data availability

The RNA sequencing data that support the findings of this study are openly available in SRA under project number PRJNA949436.

# Peer review information

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

## Acknowledgements

We are grateful to Kayi Li (Baylor College of Medicine, Houston) for the *Mcph1* KO mice. We thank Melina Devos for animal care and Zsolt Csaba for the help in using imageJ. We thank Fabien Guimiot for helping us to generate the histology images. This work was supported by grant from the Direction générale de l'offre de soins (DGOS) PHRC – MICROFANC – P100128.

## Author contributions

**Yoann Vial**: Conceptualization; Investigation; Methodology; Writing—original draft; Writing—review and editing. **Jeannette Nardelli**: Resources; Investigation; Writing—review and editing. **Adeline A Bonnard**: Investigation; Writing—review and editing. **Justine Rousselot**: Investigation. **Michèle Souyri**: Methodology; Writing—review and editing. **Pierre Gressens**: Supervision. **Hélène Cavé**: Supervision; Funding acquisition; Writing—original draft; Writing—review and editing. **Séverine Drunat**: Conceptualization; Supervision; Funding acquisition; Methodology; Writing—original draft; Writing—review and editing.

## Disclosure and competing interests statement

The authors declare no competing interests.

# Expanded View Figures

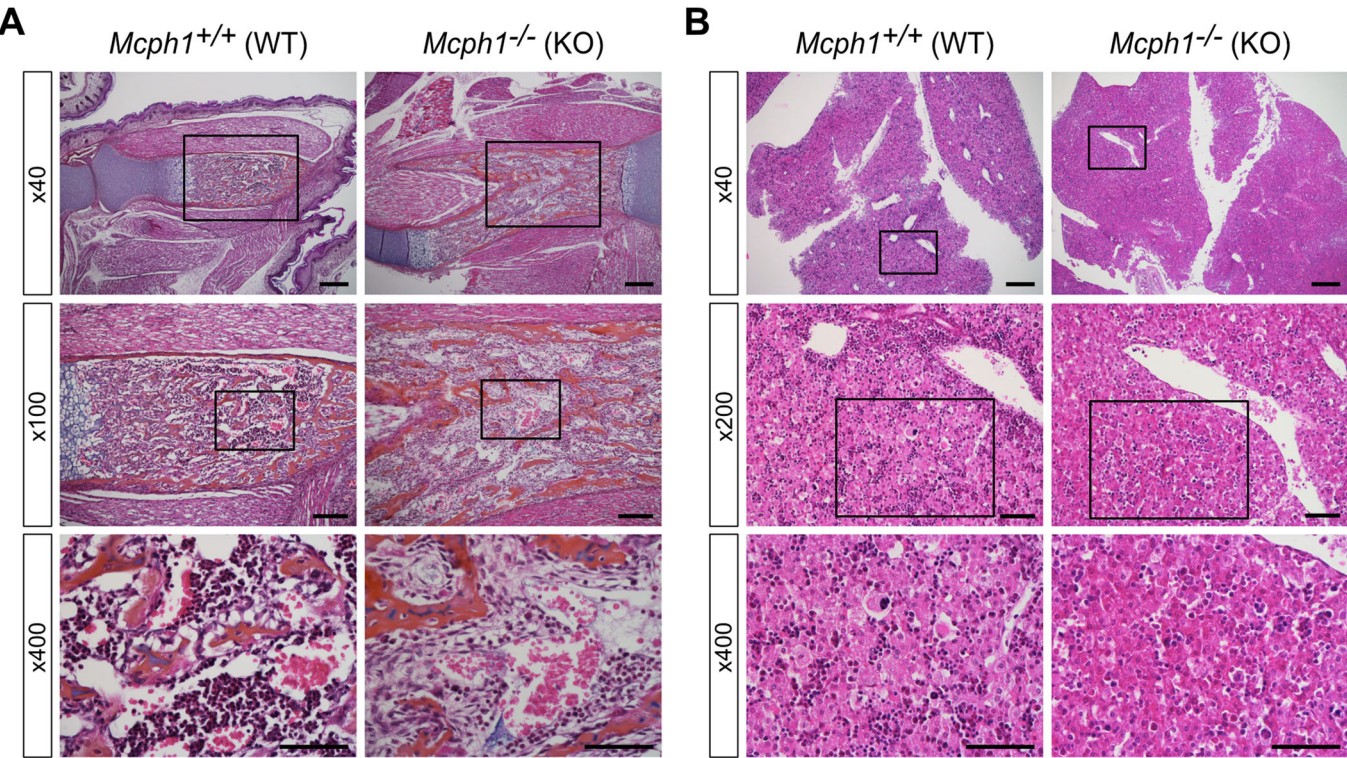

**Figure EV1.  Hematopoiesis at birth.**

Liver and femur from Wild type (WT, *Mcph1*$^{+/+}$) and knockout (KO, *Mcph1*$^{-/-}$) mice were collected at birth (day 0). Hematoxylin and Eosin staining was used to evaluate cellularity on (**A**) femur cross section ×40 (Scale bar = 200 μm), ×100 (Scale bar = 100 μm), ×400 (Scale bar = 50 μm) and (**B**) liver section ×40 (Scale bar = 200 μm), ×200 (Scale bar = 50 μm), ×400 (Scale bar = 50 μm).

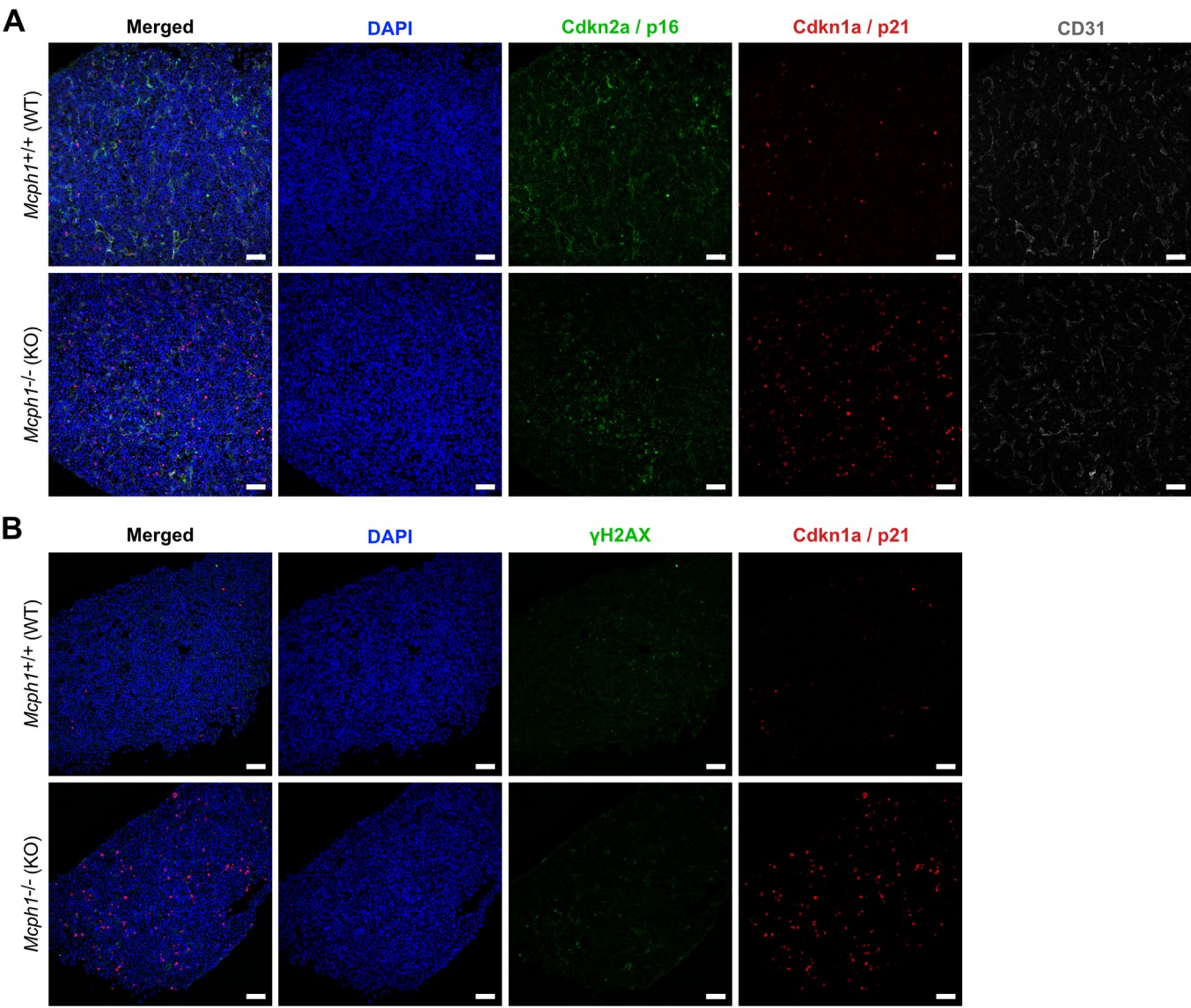

**Figure EV2. p21 is overexpressed in fetal liver in the absence of DNA damage and senescence.**

Fetal liver sections were obtained from E13.5 embryos of wild type (WT, $Mcph1^{+/+}$) and knockout (KO, $Mcph1^{-/-}$) mice. (**A**) DAPI-staining (blue) and co-immunofluorescence of p21 (red), p16 (green) and CD31 (gray). (**B**) DAPI-staining (blue) and co-immunofluorescence of p21 (red) and γH2AX (green). Data information: In (**A** and **B**), the scale bars represent 50 μm.

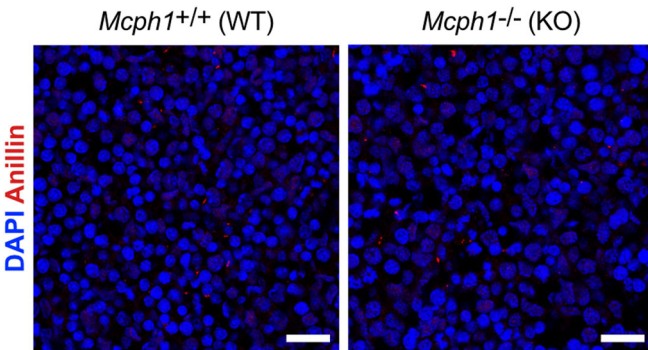

**Figure EV3.  Anillin, a protein required for the faithfulness of cytokinesis is normally expressed in fetal liver sections.**

Fetal liver sections were obtained from E13.5 embryos of wild type (WT, $Mcph1^{+/+}$) and knockout (KO, $Mcph1^{-/-}$) mice. DAPI-staining (blue) and immunofluorescence of Anillin (red). Scale bars = 20 μm.

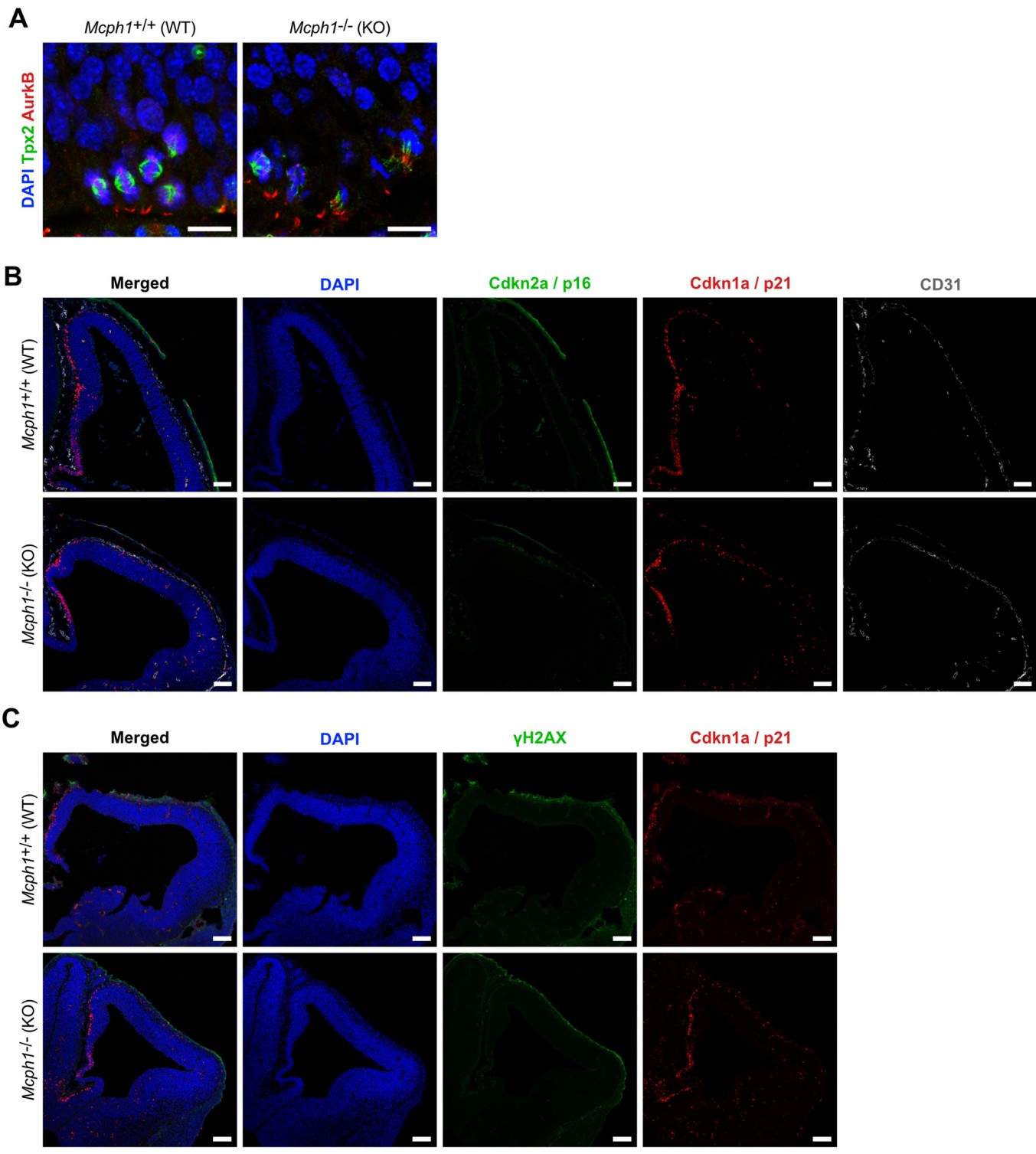

**Figure EV4.  Cell Division Defect and p21 Overexpression in Fetal Brain Revealed by Immunofluorescence.**

Coronal sections were obtained from E12.5 wild type (WT, $Mcph1^{+/+}$) and knockout (KO, $Mcph1^{-/-}$) mouse embryonic cortex. (**A**) DAPI-staining (blue) and co-IF of Aurora kinase B (AurkB, red) and Tpx2 (green). Scale bar = 10 μm. (**B**) DAPI-staining (blue) and co-immunofluorescence of p21 (red), p16 (green) and CD31 (gray). A magnified view of these images without the p16 layer is shown in Fig. 6E. Scale bar = 100 μm. (**C**) DAPI-staining (blue) and co-immunofluorescence of p21 (red) and γH2AX (green). Scale bar = 100 μm.

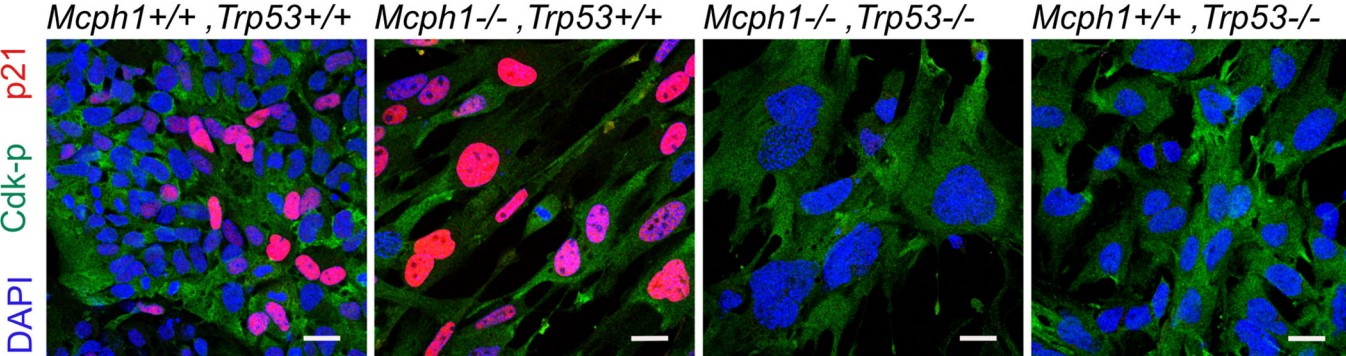

**Figure EV5.  p21 expression is mediated by p53 in NPC.**

Primary cultures of NPCs isolated from $Mcph1^{+/+},Trp53^{+/+}$ ; $Mcph1^{+/+},Trp53^{-/-}$ ; $Mcph1^{-/-},Trp53^{+/+}$ and $Mcph1^{-/-},Trp53^{-/-}$ embryos were stained with DAPI and incubated with p21 and P-Cdk antibodies. Scale bar = 10 μm.

