## [Peer Review File · EMBO Reports]

McpH1, mutated in primary microcephaly, is also crucial for erythropoiesis

Yoann Vial, Jeannette Nardelli, Adeline Bonnard, Justine Rousselot, Michèle Souyri, Pierre GRESSENS, H el ene Cav e, and Severine Drunat

Corresponding author(s): Severine Drunat (severine.drunat@aphp.fr)

Review Timeline:

Submission Date:	30th Aug 23
Editorial Decision:	17th Oct 23
Revision Received:	18th Jan 24
Editorial Decision:	22nd Feb 24
Revision Received:	28th Feb 24
Accepted:	12th Mar 24

Editor: Deniz Senyilmaz Tiebe

Transaction Report:

Dear Dr. Drunat,

Thank you for the submission of your research manuscript to our journal, which was now seen by three referees, whose reports are copied below.

The referees express interest in the proposed role of Mcph1 in regulation of cytokinesis during neurogenesis and erythropoiesis. However, they also raise significant concerns that need to be addressed to consider publication here.

In particular, both referees #1 and #2 find that the proposed link between Mcph1 and cytokinesis requires additional support and the effects of Mcph1 on genomic stability need to be better investigated (referee #1, stand first, points 3 and 4; referee #2, stand first, points 1, 2, 4, 5). Moreover, phrases regarding cytokinesis and neurogenesis need to be toned down where necessary. While we appreciate the comments of referee #3, we still find the study potentially interesting.

Should you be able to address the concerns of referees #1 and #2 fully, we would like to invite you to submit a revised manuscript. Please revise your manuscript with the understanding that the referee concerns (as in their reports) must be fully addressed and their suggestions taken on board. Please address all referee concerns in a complete point-by-point response. Acceptance of the manuscript will depend on a positive outcome of a second round of review. It is EMBO reports policy to allow a single round of major experimental revision only and acceptance or rejection of the manuscript will therefore depend on the completeness of your responses included in the next, final version of the manuscript.

We realize that it is difficult to revise to a specific deadline. In the interest of protecting the conceptual advance provided by the work, we recommend a revision within 3 months. Please discuss the revision progress ahead of this time with me if you require more time to complete the revisions, or if you have questions or comments regarding the revision (also by video chat).

1. A data availability section providing access to data deposited in public databases is missing (where applicable).
2. Your manuscript contains statistics and error bars based on $n=2$. Please use scatter plots in these cases.

You can submit the revision either as a Scientific Report or as a Research Article. For Scientific Reports, the revised manuscript can contain up to 5 main figures and 5 Expanded View figures, and it should not exceed 27000 characters. If the revision leads to a manuscript with more than 5 main figures it will be published as a Research Article. In this case the Results and Discussion section should be separate. If a Scientific Report is submitted, these sections have to be combined. This will help to shorten the manuscript text by eliminating some redundancy that is inevitable when discussing the same experiments twice. In either case, all materials and methods should be included in the main manuscript file.

3) We replaced Supplementary Information with Expanded View (EV) Figures and Tables that are collapsible/expandable online. A maximum of 5 EV Figures can be typeset. EV Figures should be cited as 'Figure EV1, Figure EV2' etc... in the text and their respective legends should be included in the main text after the legends of regular figures.

4) a .docx formatted letter INCLUDING the reviewers' reports and your detailed point-by-point responses to their comments. As part of the EMBO publication's Transparent Editorial Process, EMBO reports publishes online a Review Process File (RPF) to

accompany accepted manuscripts. This File will be published in conjunction with your paper and will include the referee reports, your point-by-point response and all pertinent correspondence relating to the manuscript.

<https://www.embopress.org/page/journal/14693178/authorguide#transparentprocess>

5) a complete author checklist, which you can download from our author guidelines

<https://www.embopress.org/page/journal/14693178/authorguide>. Please insert information in the checklist that is also reflected in the manuscript. The completed author checklist will also be part of the RPF.

6) Please note that all corresponding authors are required to supply an ORCID ID for their name upon submission of a revised manuscript (<<https://orcid.org/>>). Please find instructions on how to link your ORCID ID to your account in our manuscript tracking system in our Author guidelines

<<https://www.embopress.org/page/journal/14693178/authorguide#authorshipguidelines>>

Additional information on source data and instruction on how to label the files are available:

<https://www.embopress.org/page/journal/14693178/authorguide#sourcedata>

9) Our journal encourages inclusion of *data citations in the reference list* to directly cite datasets that were re-used and obtained from public databases. Data citations in the article text are distinct from normal bibliographical citations and should directly link to the database records from which the data can be accessed. In the main text, data citations are formatted as follows: "Data ref: Smith et al, 2001" or "Data ref: NCBI Sequence Read Archive PRJNA342805, 2017". In the Reference list, data citations must be labeled with "[DATASET]". A data reference must provide the database name, accession number/identifiers and a resolvable link to the landing page from which the data can be accessed at the end of the reference. Further instructions are available at <http://www.embopress.org/page/journal/14693178/authorguide#referencesformat>

10) Regarding data quantification (see Figure Legends:

<https://www.embopress.org/page/journal/14693178/authorguide#figureformat>)

11) The journal requires a statement specifying whether or not authors have competing interests (defined as all potential or actual interests that could be perceived to influence the presentation or interpretation of an article). In case of competing

interests, this must be specified in your disclosure statement. Further information: <https://www.embopress.org/competing-interests>

12) Please also note our reference format:

I look forward to seeing a revised version of your manuscript when it is ready. Please let me know if you have questions or comments regarding the revision.

Kind regards,

Deniz Senyilmaz Tiebe

Deniz Senyilmaz Tiebe, PhD
Editor
EMBO Reports

Referee #1:

It is known that microcephaly is a frequent feature in inherited bone marrow failure syndromes, raising questions about shared mechanisms between neurogenesis and hematopoiesis. The authors investigated *Mcph1*, a microcephaly gene, in hematological development. *Mcph1*-knockout mice displayed congenital macrocytic anemia due to impaired erythroid differentiation during fetal development. Cytokinesis problems were identified as the root cause, with tetraploid erythroid progenitors and p53 pathway activation in *Mcph1*-deficient cells. Interestingly, *Mcph1*-knockout mice also showed hypertrophied neuroprogenitors in the fetal brain with p21 overexpression, indicating a shared mechanism. However, inhibiting p53 did not reverse anemia or microcephaly, suggesting that p53 activation resulted from the proliferation defect. These findings highlight *Mcph1*'s role in fetal hematopoietic development and emphasize the impact of cytokinesis disruption on neurogenesis and erythropoiesis. From the point of neurobiologist's view, their work looks interesting, and the manuscript is considered to be acceptable after addressing the following issues. However, if the authors cannot add supportive data for the impairment of neurogenesis in the knockout mice, then the title should be changed to reflect the entire manuscript properly.

Major point:

- 1) The expression of *Mcph1* in the brain should be indicated as histological data, ideally at mRNA and protein levels, during several stages in brain development. These data are the basis for all the following analyses. If some public database could provide appropriate evidence, the authors can reuse them.
- 2) The expression of downstream molecules of *Mcph1* as well as p53-related pathway should be indicated *in vivo*. These include not only p21 but also gammaH2AX, Rho, Cit-K, etc. In addition, is the reason of thinner cortex due to apoptosis? If so, please provide the evidence. Just showing names of the pathways seems to be not much evidence to support the idea of the authors.
- 3) If the cytokinesis is impaired, how about the cell-division plane of the apical progenitor cells?
- 4) When the impaired cytokinesis starts to occur during corticogenesis, for example?

Referee #2:

In this manuscript, the authors demonstrate that mice deficient for the primary recessive microcephaly gene *Mcph1* do not only exhibit microcephaly but also severe congenital anemia, a phenotype that has not been associated with *Mcph1* mutations previously. They propose that *Mcph1* loss results in cytokinesis defects, leading to increased ploidy and aberrant terminal erythroid differentiation. Intriguingly, an increase in nuclear size/area is also seen in neural progenitors from same knockouts. Altogether, these are very interesting and potentially important observations that would appeal to EMBO Report's broad readership.

The data presented is of good quality and convincing, with adequate number of experimental repeats performed and appropriate statistical analysis employed. The degree of anemia in the knockouts is striking and from the data it is clear that a defect occurs during terminal erythroid differentiation prior to enucleation (massive decrease in Ter119 high/CD71 low erythroblasts). Furthermore, the increase in tetraploid and polyploid cells in the knockout along with the lack of rescue by p53 loss are well demonstrated.

The main weakness of the paper is the tenuous link between MCPH1 function and cytokinesis. Whereas to my knowledge MCPH1 has not been shown to contribute to cytokinesis, it has a well-established role in genomic stability, DNA damage signalling and chromosome condensation; in fact, a unifying feature of cells derived from MCPH1 patients is premature

chromosome condensation (PCC), a defect arising due to loss of inhibition of condensin II by MCPH1. PCC is not even mentioned in the manuscript. Do the authors see PCC in affected erythroblasts and at all differentiation stages? Do polyploid cells show PCC? This is not to say that loss of MCPH1 does not cause defective cytokinesis but readers will need more evidence than a couple of images of binucleate-appearing cells. As a minimum, we would need to know the frequency of binucleated cells in the population, but it would be even better if the authors could demonstrate abnormal cytokinesis in cells derived from these MCPH1 knockout mice using time-lapse microscopy. Polyploidy can also arise from endoreduplication cycles, cell cycle arrest in G2 or abortive mitosis.

Specific points:

1. Is there any DNA damage in MCPH1 knockout erythroblasts? If so, does this trigger the corresponding checkpoint?
2. Figure 4E: micronucleus is really obvious but in the example of the binucleated cell, one of the two nuclei seem to contain condensed chromatin and nuclear envelope is rough. It is difficult to see how the two nuclei would be at different cell cycle stages in shared cytoplasm. Are there other examples? Could the authors score number of these binucleated cells in population?
3. Figure 4A suggests only minor issues in cell cycle distribution in the mutant at S1 and S2, with 4D revealing more profound defects in the S3 population. It would be useful to depict the data on S3 from 4B in a similar format to 4A. With the high p21 levels, one expects most tetraploid and polyploid cells to exit from cell cycle and accumulate in G1, which is in their case is a 4N DNA content but with no EdU incorporation.
4. In Figure 6C, the authors highlight presence of a few very large p21-positive nuclei in primary neural stem cell cultures derived from knockout mice (although a couple are visible in the wild-type as well). They propose that these are polyploid cells that arise due to cytokinesis failure, and even mark a binucleated cell with white arrow. I agree that these cells are very large but to me they also appear flat and show a different morphology from other cells in the field, so I wonder if these cells are entering senescence. Have the authors tried a senescence marker? Do these cells have DNA damage?
5. In my view claims about cytokinesis failure should be toned down throughout the manuscript (especially in title and abstract) unless further experimental data can be obtained in support of this possibility. NSC cultures would allow the authors to perform timelapse microscopy and follow cells live through their cell cycle and visualise cytokinetic defects if any. An alternative approach would be (if the material is available) to stain the developing cerebral cortex of the embryonic brain (E13.5-E15.5) with a focus on the ventricular zone because the number of mitotic events taking place there should enable the authors to identify cells in telophase/cytokinesis fairly easily.
6. Western blot in Fig 3G is very faint. Does p53 also get stabilised in the KO?
7. Please include reference for this statement: "MCPH1 or Microcephalin, is the first causative gene identified in microcephaly"
8. Correct Figure S4 legend: remove 'the' from 'flow cytometry in the all precursors subsets-'.
9. Use 'nuclear area' instead of 'nuclei area' in Figure 6D legend.

Referee #3:

In this manuscript, the authors investigate the impact of loss of the microcephaly-linked MCPH1 on cell division of hematopoietic precursors and cortical neural precursors. In a set of basic analyses the authors describe 1) that loss-of-MCPH1 is associated with a thinner cortex and anemia the latter being the result of impaired terminal erythroid differentiation. 2) evidence for cytokinesis defects in hematopoietic and neural precursors, 3) increased p53 activity, whose inhibition cannot rescue the anemia and microcephaly phenotype.

My main comment is that the gain in knowledge provided by the paper is little. The link of MCPH1 with cell division has previously been described, extending the finding of impaired cell division to hematopoiesis without providing deeper mechanistic insight into a) how MCPH1 regulates cell division / cytokinesis, and b) what MCPH1 does specifically in hematopoietic and neural progenitor cells as opposed to other dividing cells, represents a limited advance in knowledge.

Response to reviewers – EMBOR-2023-58085-T

We thank the reviewers for their thoughtful comments on our manuscript. A revised version incorporating new data has been uploaded with text revisions highlighted in yellow.

The revision of the manuscript has adjusted the emphasis of the data describing neurological and hematological phenotypes to better illustrate the common challenge of these two systems. In response to the reviewers' insightful recommendations, we have made the following improvements:

1. Added new examples of binucleated erythroid cells
2. Added *in vivo* data illustrating p21 expression in the developing brain, which mirrors the fetal liver data and show overexpression of this protein.
3. Added *in vivo* data of gamma-H2AX expression, demonstrating the absence of DNA breaks in hematopoietic and cortical tissues.
4. Added *in vivo* data on p16 protein expression, indicating the absence of senescence in hematopoietic and cortical tissues.

Note that our additional figures and tables are now renamed as Extended View (EV) figures and tables or Appendix in line with EMBO reporting guidelines for revised manuscripts. Our point-by-point response (red text) to the reviewer's comments follows:

Referee #1:

It is known that microcephaly is a frequent feature in inherited bone marrow failure syndromes, raising questions about shared mechanisms between neurogenesis and hematopoiesis. The authors investigated Mcph1, a microcephaly gene, in hematological development. Mcph1-knockout mice displayed congenital macrocytic anemia due to impaired erythroid differentiation during fetal development. Cytokinesis problems were identified as the root cause, with tetraploid erythroid progenitors and p53 pathway activation in Mcph1-deficient cells. Interestingly, Mcph1-knockout mice also showed hypertrophied neuroprogenitors in the fetal brain with p21 overexpression, indicating a shared mechanism. However, inhibiting p53 did not reverse anemia or microcephaly, suggesting that p53 activation resulted from the proliferation defect. These findings highlight Mcph1's role in fetal hematopoietic development and emphasize the impact of cytokinesis disruption on neurogenesis and erythropoiesis. From the point of neurobiologist's view, their work looks interesting, and the manuscript is considered to be acceptable after addressing the following issues. However, if the authors cannot add supportive data for the impairment of neurogenesis in the knockout mice, then the title should be changed to reflect the entire manuscript properly.

Major point:

1) The expression of Mcph1 in the brain should be indicated as histological data, ideally at mRNA and protein levels, during several stages in brain development. These data are the basis for all the following analyses. If some public database could provide appropriate evidence, the authors can reuse them.

We agree with the reviewer that *Mcph1* expression during brain ontogeny is essential for a comprehensive interpretation of our results.

In a previous study, our team demonstrated through RNA *in situ* hybridization that *Mcph1* exhibits expression during the initial phases of neurogenesis and that expression begins to decline from E10.5 to E13.5, with no residual expression detected in the embryonic brain at E14.5 (Journiac N, *et al.* Cell Rep. 2020. PMID: 32294449).

We have added the following sentence to convey this information on pages 8 of the manuscript in Results - part “*Defects identified in erythroid progenitors are also found in neural progenitors*”: “McpH1 expression is prominent during the early stages of neocortical development and decline from E10.5 to E13.5, with no expression detected at E14.5 in the embryonic brain (Journiac *et al*, 2020).”

2) *The expression of downstream molecules of McpH1 as well as p53-related pathway should be indicated in vivo. These include not only p21 but also gammaH2AX, Rho, Cit-K, etc.*

In our revised manuscript, we have added immunofluorescence experiments on fetal brain and liver sections.

1) Overexpression of p21 was confirmed on the fetal brain *in vivo*, consistent with our previous results obtained on NPC from primary culture.

2) No γ H2AX foci were observed in the fetal liver and brain of KO mice, demonstrating the absence of DNA breaks in these tissues *in vivo*.

3) Fetal liver and brain sections did not show abnormal expression of Anillin and AurkB, proteins involved in the final stage of cytokinesis.

Data from fetal liver sections have been added in Figure EV2 and EV3 and commented in the text in Results part – “*Lack of McpH1 results in overexpression of the p53 target genes*”, page 6 and Results part – “*McpH1 dysfunction leads to acytokinetic mitosis in the S3 subset*”, page 7.

Data from brain sections have been added in Figure EV4 and commented in the text in Results part - “*Defects identified in erythroid progenitors are also found in neural progenitors*”, page 8.

In addition, is the reason of thinner cortex due to apoptosis? If so, please provide the evidence. Just showing names of the pathways seems to be not much evidence to support the idea of the authors.

This question is especially relevant. Indeed, our team and others have demonstrated the presence of increased caspase-3-positive cells in knockout animals, indicating that the absence of McpH1 in the developing mouse brain induces apoptosis (Journiac N, *et al*. Cell Rep. 2020. PMID: 32294449, and Zhou ZW, *et al*. DNA Repair (Amst). 2013. PMID: 23683352).

Certainly, the increased expression of p21 in the fetal brain of knockout animals that we show in this study does signify the activation of the p53 pathway. However, the knockout of p53 did not lead to the restoration of normal brain size, indicating that cortical thinning cannot be solely attributed to p53-mediated apoptosis.

Hence, we suggest that the thinner cortex is mainly due to the inability of cells to complete mitosis, resulting in reduced cell production rather than increased apoptosis. This hypothesis is further explored in the Discussion section of our manuscript on page 11.

3) *If the cytokinesis is impaired, how about the cell-division plane of the apical progenitor cells?*

This remains a key issue in MCPH, as cell division patterns profoundly influence the differentiation of apical progenitors. In particular, Zhao-Qi Wang team (Gruber R, *et al*. Nat Cell Biol. 2011, PMID: 21947081) have elegantly demonstrated that McpH1 deficiency in neuroprogenitors results in mitotic spindle misorientation, leading to altered division patterns.

Given that the primary goal of this study was to pinpoint shared disrupted pathways contributing to neurogenesis and erythropoiesis defects, and considering that cell division patterns do not impact terminal erythroid differentiation, we did not explore this issue in this particular context.

4) When the impaired cytokinesis starts to occur during corticogenesis, for example?

This question is of particular importance. Therefore, we labeled Aurora kinase B, a protein that initially localizes to prophase centromeres during the metaphase-anaphase transition and then translocates to the midbody during cytokinesis, and Tpx2, a microtubule-associated protein. We observed abnormal and disorganized mitotic spindles at the ventricular border in the neocortex of Mcph1 knockout mouse embryos at 12.5 days. The technical complexities of studying the proliferation of apical precursors before this stage prevented us from carrying out experiments at earlier stages of development. The E12.5 stage marks the onset of neurogenesis and coincides with the peak of cell death as reported in our team's publication (Journiac N, *et al.* Cell Rep. 2020. PMID: 32294449). In addition, this stage coincides with the harvest of the neuroprogenitors cultured in the present study.

Consequently, our data suggest that impairment of cytokinesis may begin before E12.5.

These data were added in Figure EV4A and commented in Results part - "*Defects identified in erythroid progenitors are also found in neural progenitors*", page 8.

Referee #2:

In this manuscript, the authors demonstrate that mice deficient for the primary recessive microcephaly gene Mcph1 do not only exhibit microcephaly but also severe congenital anemia, a phenotype that has not been associated with Mcph1 mutations previously. They propose that Mcph1 loss results in cytokinesis defects, leading to increased ploidy and aberrant terminal erythroid differentiation. Intriguingly, an increase in nuclear size/area is also seen in neural progenitors from same knockouts. Altogether, these are very interesting and potentially important observations that would appeal to EMBO Report's broad readership.

The data presented is of good quality and convincing, with adequate number of experimental repeats performed and appropriate statistical analysis employed. The degree of anemia in the knockouts is striking and from the data it is clear that a defect occurs during terminal erythroid differentiation prior to enucleation (massive decrease in Ter119 high/CD71 low erythroblasts). Furthermore, the increase in tetraploid and polyploid cells in the knockout along with the lack of rescue by p53 loss are well demonstrated.

The main weakness of the paper is the tenuous link between MCPH1 function and cytokinesis. Whereas to my knowledge MCPH1 has not been shown to contribute to cytokinesis, it has a well-established role in genomic stability, DNA damage signalling and chromosome condensation; in fact, a unifying feature of cells derived from MCPH1 patients is premature chromosome condensation (PCC), a defect arising due to loss of inhibition of condensin II by MCPH1. PCC is not even mentioned in the manuscript. Do the authors see PCC in affected erythroblasts and at all differentiation stages? Do polyploid cells show PCC?

PCC is indeed detected in lymphocytes in our mouse model, reinforcing the fidelity compared to the human phenotype. However, providing a definitive answer for erythroblasts is challenging. The progressive condensation of chromatin until terminal enucleation makes it difficult to demonstrate

PCC cytomorphologically. However, the mottled appearance of some nuclei in polyploid cells suggests that this cellular phenotype may be present in erythroblasts.

To highlight PCC in our model, Figure 1D has been enlarged to include a lymphocyte with PCC, indicated by an arrow, and the corresponding text has been added to the manuscript in Results – part “*Mcph1 null mice exhibit severe congenital anemia with impaired terminal erythroid differentiation*”, page 4: “Cytomorphologic examination of peripheral blood smears showed [...] premature chromosome condensation in lymphocytes, a cellular phenotype also observed in MCPH1 patients (Figure 1D) (Trimborn et al., 2004)”.

A new figure has been added showing the mottled appearance in polyploid erythroblasts (Figure 4D).

This is not to say that loss of MCPH1 does not cause defective cytokinesis but readers will need more evidence than a couple of images of binucleate-appearing cells.

To address this specific point we have added a figure with more examples of polyploidy erythroblasts (Figure 4D).

As a minimum, we would need to know the frequency of binucleated cells in the population, but it would be even better if the authors could demonstrate abnormal cytokinesis in cells derived from these MCPH1 knockout mice using time-lapse microscopy. Polyploidy can also arise from endoreduplication cycles, cell cycle arrest in G2 or abortive mitosis.

We have included our responses to these comments in the specific points section below.

Specific points:

1. Is there any DNA damage in MCPH1 knockout erythroblasts? If so, does this trigger the corresponding checkpoint?

This is a relevant question since the involvement of MCPH1 in DNA repair has been documented in neuroprogenitor cells. To investigate the presence of DNA breakage in erythroblasts, we performed γ -H2AX staining on fetal liver from embryos at 12.5 days of development. Our results show no evidence of DNA damage in this erythropoietic tissue.

This result is shown in Figure EV2B and commented in the text in Results part – “*Lack of Mcph1 results in overexpression of the p53 target genes*”, page6

Of note, DNA breaks were observed in neuroprogenitors only after irradiation. Under physiological conditions, there is no increase in the number of DNA breaks, as shown in the study by Zhou et al. using γ -H2AX and 53BP1 staining on *Mcph1*-ko neonatal cerebral cortex (PMID: 23683352). Since *Mcph1* is not expressed in the brain at birth, we felt it was important to extend this study by performing γ -H2AX staining on the cortex of mouse embryos at 12.5 days of development. Our results confirmed the absence of evidence of DNA damage in the developing brain (Figure EV4C).

2. Figure 4E: micronucleus is really obvious but in the example of the binucleated cell, one of the two nuclei seem to contain condensed chromatin and nuclear envelope is rough. It is difficult to see how the two nuclei would be at different cell cycle stages in shared cytoplasm. Are there other examples?

Observation of binucleated erythroblasts does indeed raise some interesting points. In the additional examples shown in Figures 4E and 4F, we show that in the majority of cases the nuclei appear similar, and indeed there are sometimes variations in their appearance. Although the hypothesis of nuclei at different stages of the cell cycle in a common cytoplasm is appealing, it is crucial to note that

cytomorphologic studies alone cannot definitively establish this hypothesis. The slightly different appearance observed may be inherent to the technique rather than representing actual differences in cell cycle stages.

In conclusion, it is difficult to draw definitive conclusions on the basis of this study alone.

Could the authors score number of these binucleated cells in population?

Experiments we have performed on the livers of newborn mice show that 12.6% of cells in the S3 sub-population have a DNA content greater than 4n. Considering that this population represents 70.7% of erythroblasts, we can say that more than 8.9% of erythroblasts are tetraploid. It's important to note that this calculation excludes binucleated erythroblasts arrested in G1.

These cytometry data are shown in Figure 1J and Figure 4D. In order to make it clearer, we have added this number in the Results section "*Mcph1* dysfunction leads to acytokinetic mitosis in the S3 subset", page 7.

3. Figure 4A suggests only minor issues in cell cycle distribution in the mutant at S1 and S2, with 4D revealing more profound defects in the S3 population. It would be useful to depict the data on S3 from 4B in a similar format to 4A. With the high p21 levels, one expects most tetraploid and polyploid cells to exit from cell cycle and accumulate in G1, which is in their case is a 4N DNA content but with no EdU incorporation.

As the reviewer rightly mentions, we can't distinguish between diploid cells in G2/M and tetraploid cells in G1 because they both have 4n DNA content. This prevented us from calculating the percentage of cells in each phase of the cell cycle.

However, to clarify this point, we have added some frames to Figure 4B that clearly indicate diploid and tetraploid cell cycle stages.

In addition, to facilitate comparison between the different erythroid subpopulations, we added the data for S0, S1, and S2 to those of S3 in Figure 4D.

4. In Figure 6C, the authors highlight presence of a few very large p21-positive nuclei in primary neural stem cell cultures derived from knockout mice (although a couple are visible in the wild-type as well). They propose that these are polyploid cells that arise due to cytokinesis failure, and even mark a binucleated cell with white arrow. I agree that these cells are very large but to me they also appear flat and show a different morphology from other cells in the field, so I wonder if these cells are entering senescence. Have the authors tried a senescence marker? Do these cells have DNA damage?

To answer this question, we performed p21, p16 and γ H2AX immunofluorescence experiments on fetal liver and brain.

An increase in p21 expression was observed in cells of both tissues of KO mice. However, p21 expression was not associated with either γ H2AX foci DNA damage or p16 expression indicative of senescence.

For fetal liver sections, data were added in Figure EV2 and commented in the text in Results part – "*Lack of Mcph1* results in overexpression of the p53 target genes", page 6.

For brain sections, data were added in Figure EV4 and commented in the text in Results part – "*Defects identified in erythroid progenitors are also found in neural progenitors*", page 8.

5. In my view claims about cytokinesis failure should be toned down throughout the manuscript (especially in title and abstract) unless further experimental data can be obtained in support of this possibility. NSC cultures would allow the authors to perform timelapse microscopy and follow cells live through their cell cycle and visualise cytokinetic defects if any. An alternative approach would be (if the material is available) to stain the developing cerebral cortex of the embryonic brain (E13.5-E15.5) with a focus on the ventricular zone because the number of mitotic events taking place there should enable the authors to identify cells in telophase/cytokinesis fairly easily.

We agree with the reviewer that although we show acytokinetic mitosis, this is not definitive evidence of cytokinesis impairment. As suggested by the reviewer, to provide further evidence of cytokinesis abnormalities, we performed Aurkb labeling on brain sections at E12.5, a protein present in the midbody prior to abscission, with a focus on the ventricular zone. Although we confirmed mitotic abnormalities, we felt that this experiment did not provide additional evidence of cytokinesis impairment.

Accordingly, and in line with the reviewer's suggestion, we have toned down the use of the term "cytokinesis defect" throughout the manuscript, including the title and abstract.

6. Western blot in Fig 3G is very faint.

Protein quantification in Figure 3G was performed by capillary electrophoresis on WES (protein simple). The Western blot image is an extrapolation of the chemiluminescence intensity.

Because the Western blot image is more accessible to the scientific community, we have retained this figure without modification.

For your information here are the raw data that clearly illustrate p21 expression in KO livers (blue curve), whereas it is notably absent in WT livers (green curve).

P21-Gapdh-Mcph1:

Actin:

Does p53 also get stabilised in the KO?

We do not have data to confirm this hypothesis. Nevertheless, our results indicate that in the absence of p53, p21 expression is abolished, but the cellular phenotype persists and is even exacerbated.

These results suggest that if p53 stabilization occurs in response to cell polyploidization to slow down the cell cycle, this response is not responsible for the observed abnormalities.

7. Please include reference for this statement: "MCPH1 or Microcephalin, is the first causative gene identified in microcephaly"

A reference was added (Kristofova M, *et al.*, *Cells*. 2022. PMID: 35053391)

8. Correct Figure S4 legend: remove 'the' from 'flow cytometry in the all precursors subsets'.

Modified. Figure S4 is now Appendix Figure 4.

9. Use 'nuclear area' instead of 'nuclei area' in Figure 6D legend.

Modified in Figure 6 and 7

Dear Dr. Drunat,

Thank you for submitting your revised manuscript. It has now been seen by two of the original referees.

As you can see, the referees find that the study is significantly improved during revision and recommend publication. However, I need you to address the points below before I can accept the manuscript.

- Please discuss PMID: 35678476 in more detail in the context of p53 rescues, either in the introduction or discussion sections.
- Please address the remaining minor concern of referee #2.
- We can accommodate up to 5 keywords. Therefore, please remove one of the keywords.
- Please remove the Author Contributions section from the manuscript.
- Please double check whether the funding information in the manuscript text matches the manuscript submission systems. We note that there may be a mismatch in the funder name.
- We note that Appendix Figure S4 is currently not called out in the text.
- The Appendix file needs a Table of Contents with page numbers. The nomenclature and callouts should be updated as Appendix Figure S1 etc. and Appendix Table S1. Please submit the Reagents table separately in the word format with in the Reagents table file format.
- Please remove the The Paper Explained section, as it is not allowed as per EMBO Reports format.
- During our routine figure checks we noted the potential re-use of section of cell between Figure 6E Mcph1^{-/-} and Figure EV4B Mcph1^{-/-} (KO) Merged., which is only allowed if the figures are derived from the same experiment, in which case it should be clearly detailed in the figure legends.
- Please add highlight boxes for. (A) femur cross section x40, x100, x400 and (B) liver section x40, x200, x400 of Figure EV1 A and B.
- Our production/data editors have asked you to clarify several points in the figure legends:
 - o Please note that a separate 'Data Information' section is required in the legends of figures EV 2a-b.
 - o Please indicate the statistical test used for data analysis in the legends of figures 3c, e; 6f-g; 8a-b.
 - o Please note that the box plot needs to be defined in terms of minima, maxima, centre, bounds of box and whiskers, and percentile in the legend of figure 6h.
 - o Please note that information related to n is missing in the legends of figures 3f; 6b; 6h; 7a.
 - o Please note that the error bars are not defined in the legends of figures 6b; 7a.
 - o Please note that scale bar and its definition are missing for figure 1d; EV 1a-b.
 - o Please note that the data citation (Data ref: He et al, 2020) does not refer to deposited experimental data, but refers to journal article. (please see <https://www.embopress.org/page/journal/14693178/authorguide#referencesformat> for data citation examples.)
- Please add the link <https://www.ncbi.nlm.nih.gov/sra/?term=PRJNA949436> to the Data Availability section, which directly resolves to the datasets.
- Papers published in EMBO Reports include a 'synopsis' and 'bullet points' to further enhance discoverability. Both are displayed on the html version of the paper and are freely accessible to all readers. The synopsis includes a short standfirst summarizing the study in 1 or 2 sentences (max 35 words) that summarize the paper and are provided by the authors and streamlined by the handling editor. I would therefore ask you to include your synopsis blurb and 3-5 bullet points listing the key experimental findings.
- In addition, please provide an image for the synopsis. This image should provide a rapid overview of the question addressed in the study but still needs to be kept fairly modest since the image size cannot exceed 550 (width) x 300-600 (height) pixels.

Thank you again for giving us to consider your manuscript for EMBO Reports, I look forward to your minor revision.

Kind regards,

Deniz Senyilmaz Tiebe

--

Deniz Senyilmaz Tiebe, PhD
Editor
EMBO Reports

Referee #1:

The manuscript has been properly revised by adding more supportive evidence. Therefore, this reviewer is fine with the revision and suggests the manuscript be published in EMBO Reports.

Referee #2:

The authors have address my comments in satisfactory manner. I would recommend the authors to consider changes to the title because the claim that Acytokinetic mitosis drives the blood and brain phenotypes of MCPH1 knockout mice is not formally proven in the paper. I agree that the presence of these abnormal mitoses could explain the phenotype of the mice but the title implies that it is the acytokinetic mitosis rather than condensation defects that cause the phenotypes, and this is not demonstrated experimentally. I would suggest a title that highlights this exciting new role of MCPH1 in erythropoiesis.

Response to the editor – EMBOR-2023-58085V3

We thank the editor for her careful comments on our manuscript. We have uploaded a revised version that incorporates the requested changes. The text revisions are highlighted in yellow in the manuscript. Our point-by-point response (red text) to the comments follows:

- Please discuss PMID: 35678476 in more detail in the context of p53 rescues, either in the introduction or discussion sections.

In our revised manuscript, we have added the following sentences in the discussion section, page 7: “This is consistent with Tátrai's study showing that loss of *Cdk5rap2*, another MCPH-associated gene, induces the formation of tetraploid erythroblasts. Although p53 is activated in this model as well, it is not responsible for the macrocytic anemia that is observed. (Tátrai & Gergely, 2022).”

- Please address the remaining minor concern of referee #2.

Title has been changed to “*Mcp1*, mutated in primary microcephaly, is also crucial for erythropoiesis”

- We can accommodate up to 5 keywords. Therefore, please remove one of the keywords.

The word "polyploidy" has been removed and the keyword "anemia" have been replaced by "Congenital anemia".

- Please remove the Author Contributions section from the manuscript.

Done

- Please double check whether the funding information in the manuscript text matches the manuscript submission systems. We note that there may be a mismatch in the funder name.

Funding information has been homogenized between the text of the manuscript and the manuscript submission system

- We note that Appendix Figure S4 is currently not called out in the text.

Figure S4 in the Appendix has been deleted. The data initially shown in Figure S4 were incorporated into Figure 4D in the previous revision of the manuscript.

- The Appendix file needs a Table of Contents with page numbers. The nomenclature and callouts should be updated as Appendix Figure S1 etc. and Appendix Table S1. Please submit the Reagents table separately in the word format with in the Reagents table file format.

A table of contents has been added and nomenclature and callouts have been updated. A new .docx file has been added that contains the Reagents table in the appropriate format.

- Please remove the The Paper Explained section, as it is not allowed as per EMBO Reports format.

Done

- During our routine figure checks we noted the potential re-use of section of cell between Figure 6E Mcph1^{-/-} and Figure EV4B Mcph1^{-/-} (KO) Merged., which is only allowed if the figures are derived from the same experiment, in which case it should be clearly detailed in the figure legends.

Figures are derived from the same experiment and this has been clearly indicated in the legends for Figure 6E, page 25, and Figure EV4B, page 26.

- Please add highlight boxes for. (A) femur cross section x40, x100, x400 and (B) liver section x40, x200, x400 of Figure EV1 A and B.

Highlight boxes have been added to the figures. For the liver section, in order to position the highlighted boxes, the images shown in the figure have been replaced by new images.

- Our production/data editors have asked you to clarify several points in the figure legends:

All of these changes have been made.

- o Please note that a separate 'Data Information' section is required in the legends of figures EV 2a-b.

Done

- o Please indicate the statistical test used for data analysis in the legends of figures 3c, e; 6f-g; 8a-b.

The statistical analyses used in our study are standard methods commonly used in RNA-seq analysis.

We now explicitly state the algorithms used to generate the statistical data in the figure legends.

Detailed descriptions of all pipelines used to generate the volcano plot, perform differential expression analysis and enrichment analysis (using GSEA or Enrichr) are provided in the Methods section. References to the original publications explaining these statistical methods are included in this section. In addition, the software tools used and their respective versions are listed in the Reagents table.

- o Please note that the box plot needs to be defined in terms of minima, maxima, centre, bounds of box and whiskers, and percentile in the legend of figure 6h. Done

- o Please note that information related to n is missing in the legends of figures 3f; 6b; 6h; 7a. Done

- o Please note that the error bars are not defined in the legends of figures 6b; 7a. Done

- o Please note that scale bar and its definition are missing for figure 1d; EV 1a-b. Done

- o Please note that the data citation (Data ref: He et al, 2020) does not refer to deposited experimental data, but refers to journal article. Done

- Please add the link <https://www.ncbi.nlm.nih.gov/sra/?term=PRJNA949436> to the Data Availability section, which directly resolves to the datasets. Done

- Papers published in EMBO Reports include a 'synopsis' and 'bullet points' to further enhance discoverability. Both are displayed on the html version of the paper and are freely accessible to all

readers. The synopsis includes a short standfirst summarizing the study in 1 or 2 sentences (max 35 words) that summarize the paper and are provided by the authors and streamlined by the handling editor. I would therefore ask you to include your synopsis blurb and 3-5 bullet points listing the key experimental findings.

Below is the summary and proposed bullet points:

Mcp1, known to be required for proper neurogenesis, also appears to be essential for erythropoiesis. Biallelic loss of *Mcp1* results in a p53-independent disruption of mitosis, ultimately leading to microcephaly and severe congenital anemia.

- *Mcp1* deficiency leads to congenital dyserythropoietic anemia in mice.
 - Loss of *Mcp1* in erythroid precursors induces acytokinetic mitosis during differentiation.
 - Acytokinetic mitosis also occurs in neural progenitors in the absence of *Mcp1*.
 - Loss of *Mcp1* leads to p53 activation in both erythroid and neuroprogenitors cells.
 - p53 inactivation fails to reverse anemia and microcephaly
- In addition, please provide an image for the synopsis. This image should provide a rapid overview of the question addressed in the study but still needs to be kept fairly modest since the image size cannot exceed 550 (width) x 300-600 (height) pixels.

A graphical abstract have been added

--

Referee #1:

The manuscript has been properly revised by adding more supportive evidence. Therefore, this reviewer is fine with the revision and suggests the manuscript be published in EMBO Reports.

Referee #2:

The authors have address my comments in satisfactory manner. I would recommend the authors to consider changes to the title because the claim that Acytokinetic mitosis drives the blood and brain phenotypes of MCPH1 knockout mice is not formally proven in the paper. I agree that the presence of these abnormal mitoses could explain the phenotype of the mice but the title implies that it is the acytokinetic mitosis rather than condensation defects that cause the phenotypes, and this is not demonstrated experimentally. I would suggest a title that highlights this exciting new role of MCPH1 in erythropoiesis.

Dear Dr. Drunat,

Thank you for submitting your revised manuscript. I have now looked at everything and all is fine. Therefore, I am very pleased to accept your manuscript for publication in EMBO Reports.

Congratulations on a nice work!

Kind regards,

Deniz Senyilmaz Tiebe

--

Deniz Senyilmaz Tiebe, PhD

Editor

EMBO Reports

--
